# Multi-Source Remote Sensing for large-scale biomass estimation in mediterranean olive orchards using GEDI LiDAR and Machine Learning

Francisco Contreras[1], María L. Cayuela[1], Miguel A. Sánchez-Monedero[1], Pedro Pérez-Cutillas[2]

[1] Department of Soil and organic waste management for greenhouse gas mitigation in agriculture, CEBAS-CSIC, Campus Universitario de Espinardo, 30100, Murcia, Spain
[2] Department of Geography, University of Murcia, C. Santo Cristo, 1, 30001, Murcia, Spain

*Correspondence to: Francisco Contreras (fcontreras@cebas.csic.es)*

**Abstract**

Accurate estimation of Above-Ground Biomass Density (AGBD) is essential for assessing carbon stocks and promoting sustainable agricultural practices. This study integrates multi-source remote sensing data, including GEDI LiDAR, optical, SAR, and topographic variables, to predict AGBD in Mediterranean olive orchards using a Random Forest regression model implemented on Google Earth Engine (GEE). The proposed volumetric approach, based on GEDI L2A canopy height and
15 dendrometric parameters, provides a scalable framework for large-scale biomass estimation using lidar technologies on satellite platforms. The model's predictive performance varied depending on data combinations, with the fully multi-source configuration achieving the most consistent results, although overall accuracy remained moderate due to sensor constraints and the inherent limitations of the proposed exploratory framework. NDBI, slope, HV polarization, and MCARI1 were identified as the most influential predictors. The spatial analysis revealed that Spain exhibited the highest total AGB stock
among the studied countries, followed by Italy and Greece, reflecting their dominance in olive production. Despite its limitations in precision at fine spatial scales, this exploratory study demonstrates the potential of integrating LiDAR, optical, and SAR data to evaluate biomass distribution in low-stature vegetation. The proposed framework offers a cost-effective and scalable strategy for large-scale carbon monitoring and supports data-driven agricultural management toward more sustainable Mediterranean production systems.

**Keywords:** Biomass Modeling, Carbon Sequestration, Vegetation Structure, Crop Monitoring, Ecosystem Services, Precision Agriculture, Remote Sensing, Geospatial Data Integration

## 1 Introduction

Olive tree groves (*Olea europaea* L.) are among the most significant crops globally, serving as a mainstay of agricultural and economic systems, particularly in Mediterranean-climate regions where conditions are highly favorable for their cultivation (Proietti et al., 2014). Beyond their economic relevance, olive groves play a crucial role in global climate dynamics due to their ecological and agronomic characteristics. As perennial plants with extensive root systems, olive trees contribute to soil carbon sequestration, effectively storing carbon in both their biomass and the soil over long periods. This natural process helps mitigate climate change by reducing the concentration of greenhouse gases in the atmosphere (Pardo et al., 2017). Therefore, integrating innovative management strategies in olive cultivation presents a promising approach to enhancing climate change resilience and promoting sustainable agricultural practices.

A fundamental parameter for assessing biomass in these ecosystems is Above-Ground Biomass Density (AGBD). The Land Product Validation Subgroup of the Committee on Earth Observation Satellites (CEOS) defines AGBD as the standing dry mass of live or dead woody matter derived from trees or shrubs, typically expressed in megagrams per hectare (Mg·ha$^{-1}$). This definition is particularly relevant for Earth Observation (EO) applications, as it encompasses both living and dead biomass. While some scientific disciplines differentiate between live wood, leaf mass, or organic debris, EO methodologies primarily focus on standing biomass. For instance, LiDAR sensors can detect both live and dead biomass components due to their structural sensitivity, whereas radar systems are typically more responsive to moisture content and object size, which often results in a stronger signal from live woody biomass. Nonetheless, SAR can also detect dead biomass, especially when structural elements persist. However, converting EO-derived measurements into biomass estimates often involves allometric models, which may inconsistently account for the proportions of live and dead wood in calibration datasets (Duncanson et al., 2021).

Despite significant advances in remote sensing, accurately estimating biomass in olive groves presents unique challenges. Forest ecosystems typically exhibit complex structures shaped by natural processes (Sinha et al., 2015), while olive orchards are generally more structurally ordered due to intentional planting arrangements. However, this apparent regularity can be misleading, as olive plantations still display substantial structural variability across sites. Differences in planting frameworks, irrigation regimes, pruning practices, and management intensities introduce considerable heterogeneity, even within this controlled agricultural context. Additionally, olive trees have a distinct architecture, with a substantial portion of their biomass concentrated in the crown, which complicates the application of generic biomass estimation models (Velázquez-Martí et al., 2014). The complex relationship between canopy cover and total biomass further increases uncertainty in satellite-derived biomass estimates (Rodríguez-Lizana et al., 2023). These factors highlight the necessity of developing specific methodologies tailored to olive groves.

To improve biomass estimation accuracy, alternative approaches have been proposed. Velázquez-Martí et al. (2011) developed a methodology that incorporates multiple allometric characteristics, including olive tree variety, tree height, and irrigation conditions, allowing for highly accurate predictions in specific crop types. Conventionally, forest inventories have been the

60 primary source of biomass data for national assessments (Hunka et al., 2025; Nesha et al., 2022), where the availability of allometric databases from diverse locations supports their implementation (Jucker et al., 2022). These methods are widely recognized for their accuracy and are extensively applied in forestry sector inventories. However, they require substantial financial resources and manual labor, making them less feasible for large-scale agricultural applications.

An alternative method estimates total biomass using geometric parameters to derive the volume of woody material in different

tree structures (Kebede and Soromessa, 2018; Velázquez-Martí et al., 2014). This approach suggests that olive trees typically allocate approximately 60% of their biomass to the crown and 40% to the stem or trunk (Brunori et al., 2017). Allometric modeling techniques, which derive tree volume for biomass estimation, have shown promising results in capturing structural variations in olive groves. While allometric models provide high accuracy in localized studies, their large-scale applicability is often constrained by the need for extensive field data collection (Brede et al., 2022).

Conversely, volumetric models represent a scalable and potentially accurate alternative for biomass estimation (Li et al., 2024). However, when applied to heterogeneous landscapes, their reliability strongly depends on the resolution and accuracy of the input data. In the case of GEDI, while it provides valuable structural information, limitations in both vertical and horizontal precision introduce considerable uncertainty. Airborne LiDAR data can provide highly accurate three-dimensional structural information but is costly and limited in temporal coverage (Dubayah et al., 2020) . Meanwhile, optical and radar satellite

sensors offer frequent observations over large areas but are constrained by spatial resolution and sensitivity to canopy characteristics. High-resolution imagery captured by Remotely Piloted Aircraft Systems (RPAS) enables detailed local measurements, often outperforming conventional satellite-based approaches in precision (Perna et al., 2024). However, RPAS-based methods lack scalability and are labor-intensive, restricting their application to regional studies (Estornell et al., 2015). The integration of multiple data sources, including LiDAR, optical, radar, and topographic datasets, has emerged as a potential

solution to mitigate these limitations (Shendryk, 2022).

Facing such a new challenge, Google Earth Engine (GEE) has emerged as a powerful cloud-based platform for geospatial analysis, enabling efficient processing of large-scale Earth observation datasets while supporting multitemporal and multiscale investigations (Gorelick et al., 2017). Its extensive repository of remote sensing imagery, coupled with high-performance computing capabilities, facilitates the harmonization of disparate datasets. This capability is particularly relevant for biomass

estimation, where the temporal variability of vegetation dynamics and the spatial heterogeneity of landscapes necessitate adaptive and scalable analytical frameworks. By leveraging GEE's capacity to process vast amounts of remote sensing data in near real-time, it is possible to track biomass fluctuations across different temporal resolutions and apply models across diverse ecological and climatic regions (Pérez-Cutillas et al., 2023).

Given the limited availability of remote sensing-based biomass estimates in olive groves, this study aims to develop a

90 methodology for estimating AGBD in olive groves through the integration of multi-source remote sensing data, including the Global Ecosystem Dynamics Investigation data (GEDI LiDAR), optical and infrared satellite imagery, synthetic aperture radar (SAR), and topographic variables. By evaluating the relative contribution of each data type, the study seeks to determine the most effective combination of remote sensing inputs for biomass estimation. Additionally, the proposed methodology will be

applied at a European scale, leveraging datasets such as Corine Land Cover (CLC) as other datasets at greater precision to ensure the application of the model on the correct land use. Ultimately, this research aims to provide an exploraty tool for monitoring carbon stocks in form of biomass in olive orchards. In the case of olive orchards, this is particularly relevant given the possibility of transforming residual biomass into stable carbon forms, enabling the development of carbon sequestration protocols, supporting sustainable agricultural management and climate change mitigation strategies.

## 2 Material and methods

### 2.1. Study area

To identify olive cultivation areas, the CLC database was used. This is a European land use dataset that includes a specific category for "Olive Groves". However, CLC has limitations in accurately delineating parcel boundaries due to its generalized spatial resolution, making it challenging to create a reliable dataset for model training. To address this limitation, the calibration analysis is conducted in Spain, utilizing the 'Sistema de Información Geográfica de Parcelas Agrícolas'. SIGPAC is a governmental agronomic management tool that provides highly detailed cadastral information on agricultural parcel boundaries. With 2,695,055 hectares of olive plantations registered in the database, SIGPAC enables a more precise selection of olive groves for model calibration. In comparison, CLC database reports 3,587,300 hectares, clearly overestimating the actual olive grove area in Spain.

Furthermore, Spain is a key reference for olive cultivation, representing approximately 41% of Europe's total olive grove area. This extensive coverage provides a robust dataset for model training and validation. Within Spain, the Mediterranean agro-climatic region is particularly relevant for olive production, characterized by warm and dry summers and mild winters (Pardo et al., 2017). In these environments, annual precipitation varies between 200 and 600 mm, with a markedly seasonal distribution. Such conditions favor drought-resistant crops such as olives, which thrive in shallow, calcareous soils under high temperatures and intense solar radiation (Deitch et al., 2017; Urdiales-Flores et al., 2024).

The extensive cultivation of olive trees in Spain has significant economic and environmental implications. Beyond being a major contributor to the regional economy, olive groves offer opportunities for biomass valorization and carbon sequestration, supporting both agricultural sustainability and bioenergy development (Rosúa and Pasadas, 2012). A key example is Jaén province, the world's leading olive oil-producing region, with 550,000 hectares of olive groves. This accounts for over 25% of Spain's total olive-growing area and 42% of Andalusia's cultivated land (Fernández-Lobato et al., 2024), making it an essential site for biomass assessment and model validation.

### 2.2. Methodological framework approach

AGBD prediction in olive orchards focuses on quantifying the biomass associated with olive trees within a gridded spatial framework. This approach involves estimating the total number of trees per pixel, the average crown diameter per pixel, and applying a volumetric modelling method to derive biomass estimates.

Currently, there are products such as GEDI L4A product that provides AGBD estimates based on plant functional types (PFTs). This product employs a zonal stratification approach, applying specific models depending on the geographic region (Kellner et al., 2023). These models are developed based on the dominant vegetation types within each area. While this method is appropriate for large-scale mapping and generalization, it becomes insufficient when focusing on specific tree species. In such cases, more detailed approaches are required to enable more accurate monitoring of biomass using remote sensing techniques.

The proposed framework derives AGBD estimates from the GEDI L2A product (Dubayah et al., 2021), which provides canopy height metrics transformed into biomass values using a volumetric model tailored to olive tree morphology, supported by the estimations of tree density and canopy cover. This dataset is integrated with remote sensing variables and analyzed to assess the relative importance of each data type in biomass estimation. GEE was employed to download, pre-process, filter, and transform the satellite images.

The workflow was structured into six main phases: (1) acquisition of remote sensing data; (2) processing of all datasets to generate derived subproducts from the original inputs; (3) modeling of AGBD using a volumetric approach; (4) application of machine learning techniques to estimate both intermediate subproducts and AGBD; (5) large-scale application of the model; and (6) internal model validation using 25% of the training dataset, and an additional independent validation using ALS-derived referenced data (Figure 1).

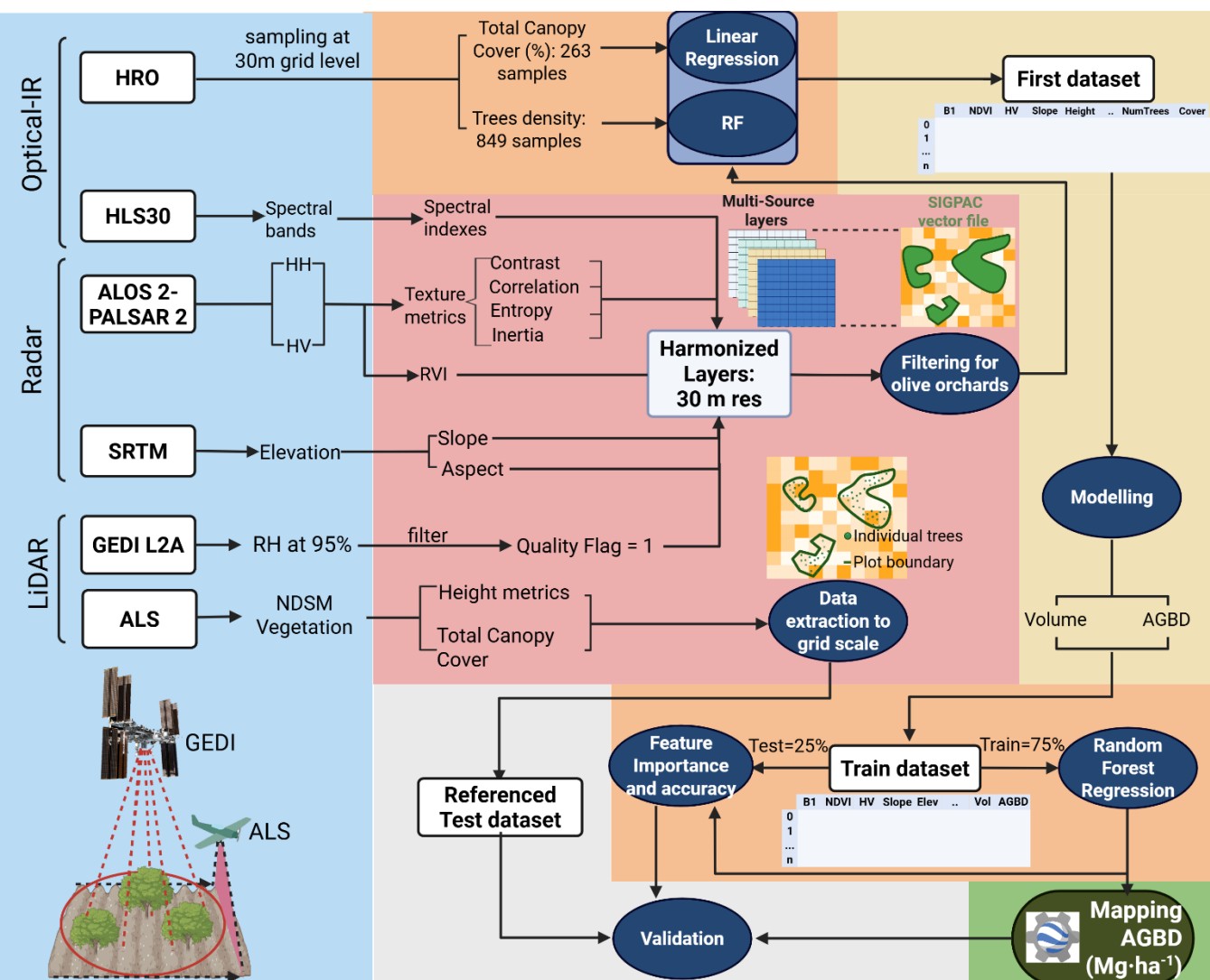

**Figure 1**. **Methodological Flowchart for AGBD mapping in Olive Orchards. The workflow comprises six sequential stages, each highlighted by a distinct color frame: data acquisition (blue frame), preprocessing (red frame), training dataset generation (yellow frame), model training (orange frame), biomass mapping (green frame), and testing (grey frame). Remote sensing data sources include GEDI (Global Ecosystem Dynamics Investigation) LiDAR onboard the ISS (International Space Station), Airbone Laser Scanning (ALS), High Resolution Orthoimagery (HRO), optical imagery (HLS: Sentinel-2, Landsat 8/9), SAR (Synthetic Aperture Radar) data (ALOS-2 PALSAR-2), and topographic information (SRTM – Shuttle Radar Topography Mission). To improve plot delineation, SIGPAC (Sistema de Información Geográfica de Parcelas Agrícolas), a Spanish Geographic Information System for Agricultural Plots is used. GEDI-derived biomass values (L2A canopy height metrics) are correlated with spectral indices and radar-derivatives. A Random Forest (RF) model is trained and applied at a large scale to generate a biomass density map.**

The methodology begins with the processing of multi-source remote sensing data, including GEDI LiDAR, optical imagery (Sentinel-2, Landsat 8/9), Synthetic Aperture Radar (SAR) from ALOS-2 PALSAR-2, and topographic data from SRTM. The

spatial distribution of AGBD across Mediterranean accuracy in olive grove delineation, SIGPAC cadastral data is used to refine the initial classification from the CLC database, that is, SIGPAC was used for the filtering of selected sites for training and testing. Training dataset construction and biomass estimation have been carried out using the filtered GEDI data that are linked to spectral, radar, and topographic variables, creating a comprehensive training dataset. L2A canopy height metrics serves as canopy height values for AGBD estimation after its processing using a volumetric approach adapted to olive cultivation.

Remote sensing data provides a continuous spatial grid, while GEDI data provides data only at certain points at the footprint level. It is for this reason that we are assessing the ability of common remotely sensed data to reliably estimate biophysical variables by integrating both types of data. The Random Forest (RF) regression model is employed to large-scale predictions in the GEE platform and to analyze the contribution of each predictor variable, optimizing biomass prediction accuracy. Lastly, model application and biomass mapping, once trained, is applied on a large spatial scale to generate a high-resolution (30 m) biomass density map for olive groves only in the areas considered "Olive Orchards" by the CLC database.

### 2.3 Data Sources and Preprocessing

### 2.3.1 GEDI LiDAR Data

GEDI is a spaceborne LiDAR instrument onboard the International Space Station (ISS) used to capture detailed vertical vegetation structure data. It operates as an active remote sensing system emitting laser pulses and analyzing the returned waveform, which encodes the vertical distribution of vegetation and ground surfaces within each footprint (Dubayah et al., 2020). This waveform enables the extraction of key structural metrics, such as canopy height, relative height percentiles (RH metrics), and ground elevation, even under dense vegetation cover (Asner et al., 2012).

Each GEDI footprint has a diameter of approximately 25 meters and is separated 60 meters apart along track, allowing for detailed sampling of vertical structure at fine spatial resolution (Duncanson et al., 2022). Also, the L2A product has a relative uncertainty associated that interferes in the sampling vertical and positional accuracy.

GEDI provides multiple data products, classified by processing levels. The L2A product offers relative height metrics (RH metrics), which describe the vertical structure of vegetation (Kellner et al., 2023). The GEDI L2A product delivers these RH metrics, which represent the height above ground below which a given percentage of the total returned energy has been accumulated, which is useful as biomass model inputs.

### 2.3.2 Optical and Infrared Data

The Harmonized Landsat-Sentinel (HLS) dataset integrates Landsat 8-9 and Sentinel-2A/B imagery to provide consistent, high-resolution optical and infrared data at a 30 m spatial resolution (Masek et al., 2021). This resolution is used as a key layer for harmonizing all multi-source layers. By merging these sensors, HLS enables global observations with a revisit frequency of 2–3 days, offering the highest temporal resolution among freely available high-resolution imagery. Sentinel-2's MultiSpectral Instrument (MSI) operates in the visible (VIS), near-infrared (NIR), and shortwave infrared (SWIR) ranges, whereas Landsat's Operational Land Imager (OLI) and Thermal Infrared Sensor (TIRS) provide coverage in the visible,

infrared, and thermal regions (Masek et al., 2018). To ensure consistency, the HLS product undergoes rigorous preprocessing, including atmospheric correction, cloud-shadow masking, illumination and view angle normalization, spatial co-registration, and spectral band harmonization (Masek et al., 2021). For the HLS data processing, all available images from the annual period between 2020 and 2022 were selected. A filtering step was then applied to exclude pixels with more than 10% cloud cover. Subsequently, a temporal median filter was applied to the remaining data. This process enables the generalization of the spectral response over the entire annual period, reducing the influence of outliers and transient atmospheric effects.

Spectral indices derived from HLS bands were computed within the Google Earth Engine (GEE) platform. The indices, detailed in **Table 1**, were extracted from the Index Database (IDB) (Henrich, 2012).

High-Resolution Orthophotography (HRO) from the Spanish National Aerial Orthophotography Plan- PNOA (https://pnoa.ign.es/web/portal/pnoa-lidar/presentacion) was also incorporated as a key optical data source in this study, which served as the basis for the sampling process to estimate the crops' planting framework and the canopy cover. These orthophotos have a spatial resolution of 0.25 meters. This dataset was used exclusively during the sampling phase, due to its high spatial resolution and its suitability for accurately guiding the sampling process.

### 2.3.3 Synthetic Aperture Radar (SAR) and Topographic Data

ALOS-2 PALSAR-2 is a Synthetic Aperture Radar (SAR) system that operates as an active remote sensing method. It emits microwave pulses and measures their backscatter to detect surface features. SAR wavelengths are long enough to penetrate cloud cover, ensuring reliable data acquisition regardless of atmospheric conditions (Shimada et al., 2014). PALSAR-2 is a highly valuable data source, as its operating frequency in the L-band allows for a more effective characterization of overall vegetation structure compared to other SAR systems operating in different frequency bands. For instance, C-band sensors offer higher spatial resolution but are more sensitive to seasonal variations (Qin et al., 2025). The objective of this study was to work with annual SAR data, making ALOS-2 PALSAR-2 the most suitable option. Moreover, the Google Earth Engine (GEE) catalog provides a ready-to-use annual mosaic generated from multiple image strips, which facilitates consistent temporal analysis and simplifies data preprocessing. ALOS-2 PALSAR-2 data were processed directly within GEE to enhance radiometric quality and correct for topographic effects. First, radiometric calibration was applied by converting the digital number (DN) values into backscatter coefficients expressed in decibels (dB) expressed in Equation 1:

$$\gamma_0 = 10 \cdot log_{10}(DN^2) - 83 \qquad (1)$$

Subsequently, topographic correction was performed using complementary SRTM data to estimate slope and aspect, allowing for the calculation of the local incidence angle. This angle was then used to adjust the backscatter values, compensating for the influence of terrain relief on SAR signal intensity. Finally, a spatial median filter with a 50-meter radius was applied to reduce speckle noise. The HH and HV polarizations were processed separately and later combined to derive subsequent SAR-based products.

Regarding topographical data, this study employed the Shuttle Radar Topography Mission (SRTM) Global 1 Arc-Second v003 Digital Elevation Model (DEM), which provides global elevation data at a 30 m spatial resolution (NASA JPL, 2013). SRTM-

derived terrain variables, including slope and aspect, were integrated to improve biomass estimation models, accounting for the influence of topography on vegetation growth patterns.

### 2.3.4. Airborne LiDAR Scanning (ALS) Data

PNOA-Normalized Digital Surface Model for vegetation (NDSM-veg) is the derived LiDAR product selected to validate the model predictions. This dataset is a 2.5-meter resolution product derived from LiDAR point clouds collected through Airborne Laser Scanning (ALS) missions. This product is already processed through interpolation of height relative to the ground, based on vegetation classes (low, medium, and tall vegetation) derived from LiDAR flights conducted during the first coverage of PNOA. This dataset was resampled at the 30-meter grid scale, allowing for the estimation of both average crown diameter and the number of trees per pixel.

**Table 1. Summary of Remote Sensing Variables and Features Employed**

| Data type | Image Data | Band | Description |
|---|---|---|---|
| LiDAR | GEDI | RH | L2A: Relative Height (RH) Metrics Percentiles |
| | | Quality Flag | L2A: Flag indicating likely invalid waveform |
| | ALS | NDMS-Veg | Normalized Digital Model Surface of the vegetation class |
| Optical-Infrared | HLS: Landsat 8/9 and Sentinel 2 | Band 1 | Coastal Aerosol |
| | | Band 2 | Blue |
| | | Band 3 | Green |
| | | Band 4 | Red |
| | | Band 5 | NIR |
| | | Band 6 | SWIR 1 |
| | | Band 7 | SWIR 2 |
| | | Band 9 | Cirrus |
| | | NDVI | (NIR-RED) / (NIR+RED) |
| | | GI | GREEN / RED |
| | | NDBI | (NIR-SWIR) / (NIR+SWIR) |
| | | GNDBI | (NIR-GREEN) / (NIR+GREEN) |
| | | MCARI1 | 1.2 * ((2.5*(NIR-RED)) - (1.3*(NIR-GREEN))) |
| | | GLI | 2*(GREEN + RED + BLUE) / 2*(GREEN - RED - BLUE) |
| | | EVI | 2.5 * ((NIR - RED) / (NIR + 6 * RED - 7.5 * BLUE + 1)) |
| | | SAVI | ((NIR - RED) / (NIR + RED + 0.5)) * 1.5 |
| | HR Orthoimagery | Cover | Total Canopy Cover (%) |
| SAR | ALOS2 PALSAR2 | HH | Polarization backscattering coefficient L-Band |
| | | HV | Polarization backscattering coefficient L-Band |
| | | Contrast | GLCM Texture: Contrast |
| | | Correlation | GLCM Texture: Correlation |

| | | Entropy | GLCM Texture: Entropy |
|---|---|---|---|
| | | Inertia | GLCM Texture: Inertia |
| | | RVI | (4 * HV) / (HH + HV) |
| Topographic | SRTM | Elevation | SRTM V3 product (SRTM Plus) |
| | | Slope | Derived by "Elevation" |
| | | Aspect | Derived by "Elevation" |

Note: Acronyms used in this table are as follows: GEDI (Global Ecosystem Dynamics Investigation), , L2A (GEDI product providing Canopy Height Metrics), RH95 (Relative Height at 95%), NDMS (Normalized Digital Model Surface), HLS (Harmonized Landsat-Sentinel), NIR (Near-Infrared), SWIR (Shortwave Infrared), NDVI (Normalized Difference Vegetation Index), NDWI (Normalized Difference Water Index), GI (Green Index), GNDBI (Green Normalized Difference Built-up Index), MCARI1 (Modified Chlorophyll Absorption Ratio Index 1), GLI (Green Leaf Index), EVI (Enhanced Vegetation Index), SAVI (Soil-Adjusted Vegetation Index), HRO-PNOA (National Aerial Orthophotography Plan), SAR (Synthetic Aperture Radar), ALOS-2 (Advanced Land Observing Satellite-2), PALSAR-2 (Phased Array type L-band Synthetic Aperture Radar-2), HH (Horizontal-Horizontal polarization), HV (Horizontal-Vertical polarization), GLCM (Gray Level Co-occurrence Matrix), RVI (Radar Vegetation Index), SRTM (Shuttle Radar Topography Mission), and DEM (Digital Elevation Model).

## 2.4 AGBD Estimation Methodology

### 2.4.1. Tree Density and Canopy Cover Modeling

Tree density per hectare and tree crown diameter are essential variables for applying the proposed approach at pixel level. A sampling strategy was implemented to obtain these variables using data collected from various points. This sampling allowed producing two derivate products that subsequently interacted with GEDI canopy top metrics to produce the volumetric estimations at 30m grid level.

For the estimation of tree crown size, a method was sought that could incorporate the presence of multiple trees within a single pixel. This enabled the calculation of the average crown diameter across all trees within a pixel, thereby allowing for the generalization of this attribute at the 30-meter-pixel level. Specifically, knowing the proportion of the pixel covered by canopy and the total number of trees within it makes it possible to reliably estimate the average crown diameter.

To this end, 263 'Total Canopy Cover' sampling points were collected across Spain olive orchards (Figure 2). To simplify the canopy cover estimation process, a simple linear regression analysis was conducted, using SAVI as the predictor variable and canopy cover as the response variable. This approach enabled the use of the resulting regression equation to estimate the approximate canopy cover for each pixel using spectral bands.

Additionally, before calculating SAVI, the Bare Soil Index (BSI) was used as an indicator to assess the type of ground cover present. This step improved the characterization of soil conditions, enhancing the reliability of SAVI-based canopy cover estimations. In this case, a threshold-based approach was applied to define the SAVI L-factor: pixels with BSI values less than 0 (indicating non-bare soil) were assigned an 'L' value of 0, while pixels with BSI values greater than 0 (indicating a significant influence of bare soil) were assigned an 'L' value of 0.5. This adjustment is particularly beneficial in Mediterranean environments, where high variability in soil cover is common. For instance, in olive groves with partial vegetative ground

cover, distinguishing between canopy and bare soil can be challenging. SAVI helps mitigate these limitations by reducing the soil background effect in such heterogeneous landscapes.

Regarding the number of trees per hectare, the sampling method involved manually counting individual trees within a 30-meter grid (Figure 2). This approach enabled the estimation of tree density at the hectare scale by applying a spatial conversion from the 30-meter grid (900 m²) to a 100-meter grid (10000 m²). The sample number of tree density was 849 across Spain, using sampling in multiple kinds of olive tree cultivation frames. To perform the estimation, Random Forest Regression was used.

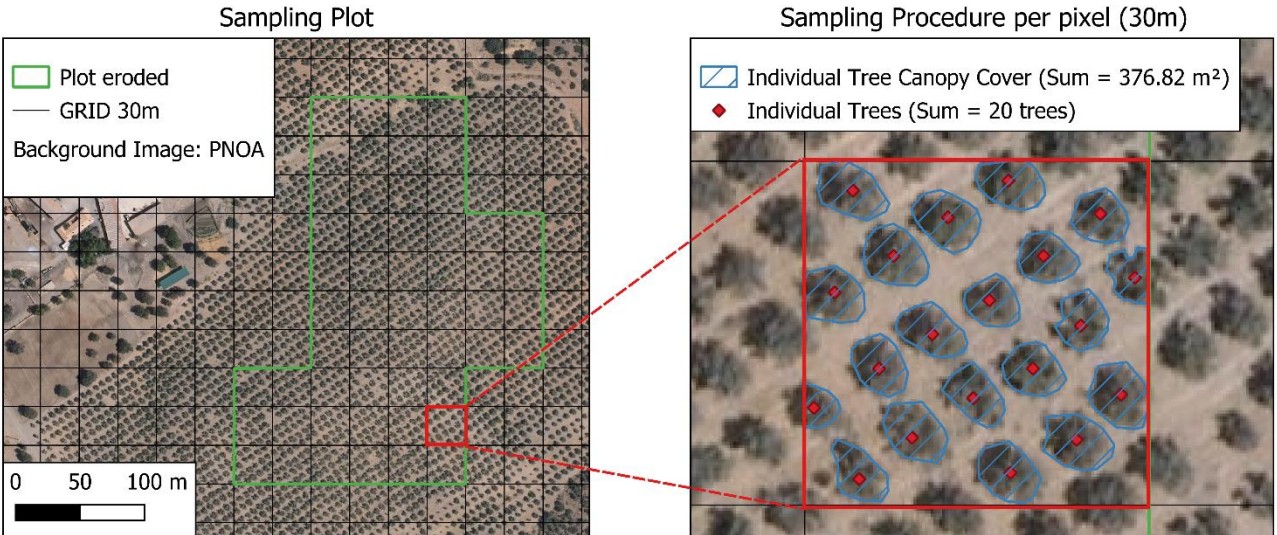

**Figure 2. Illustration of the sampling procedure implemented to estimate tree density (trees·ha⁻¹) and canopy cover. The left panel displays an example of single eroded olive grove parcel, extracted from the SIGPAC database, with a 30-meter pixel selected for sampling highlighted in red. The right panel depicts the methodological workflow applied during the sampling phase, which was used to derive the training dataset for model development.**

### 2.4.2 Theoretical Framework for Biomass Modeling

The estimation of AGBD was based on a volumetric modeling approach that combined GEDI L2A LiDAR metrics with canopy cover and tree density predictions derived from multi-source imagery. The Relative Height at 95% (RH95) from GEDI L2A was used as a proxy for canopy top height, representing tree height within each footprint. Canopy Cover was calculated to indicate the proportion of the GEDI footprint occupied by the tree canopy. To estimate crown volume, the model proposed by Velázquez-Martí et al. (2014) for olive trees, using a hemispherical volumetric model. This model converts crown volume to biomass using a wood density (WD) of 0.76 g·cm⁻³, which is consistent with values reported for olive trees (Brunori et al., 2017; Kebede and Soromessa, 2018). This WD value was applied as a constant to convert volumes into biomass units.

Crown volume (Cvol) was estimated using the crown radius (Cdiam/2) and crown height, both derived from GEDI and canopy cover metrics. The crown height was calculated from RH95, assuming that approximately 65% of the total tree height

corresponds to the crown, reflecting typical olive tree morphology (Velázquez-Martí et al., 2014). The crown diameter (Cdiam) is the average crown diameter per tree within the 30m pixel, that was obtained from the total canopy cover (Cover), pixel area (Parea), and the number of trees per pixel (Mpix), following Equation (2):

$$Cdiam = 2\sqrt{\frac{(\text{cover} \cdot \text{Parea})/\text{Mpix}}{\pi}} \qquad (2)$$

The hemispherical crown volume (Hvol) was calculated using the formula for a hemisphere (Equation 3):

$$Hvol = \frac{2}{3}\pi r^2 h \qquad (3)$$

Where, $r = \frac{C_{diam}}{2}$ is the crown radius, and $h$ is the crown height (65% of RH95).

To obtain the actual crown volume (Cvol), the hemispherical volume was adjusted by an Occupational Factor (OF) specific to olive tree crown morphology (Velázquez-Martí et al., 2014) as shown in Equation 4:

$$Cvol = \text{Hvol} \cdot OF \qquad (4)$$

The stem volume was approximated based on its relationship with crown volume, following the proportional model described by Velázquez-Martí et al. (2014). Crown and stem volumes were converted into biomass using the wood density (WD) value of 0.76 g·cm⁻³. The total AGBD was then computed by multiplying the total tree volume (Tvol, sum of crown and stem volumes) by wood density and the number of trees per hectare (Mi), as shown in Equation 4:

$$AGBD = \text{Tvol} \cdot \text{WD} \cdot \text{M}i \qquad (5)$$

Lastly, the computed AGBD values at 30m pixel level were subsequently used as training data in a RF model, which was applied across the study area to generate a spatially continuous map of AGBD for olive orchards. The RF model was trained with a combination of LiDAR-derived metrics, canopy cover from spectral indices, and topographic variables, enabling large-scale biomass estimation.

**2.4.3 Dataset Construction and Footprint Filtering**

The construction of the AGBD dataset involved integrating GEDI L2A LiDAR metrics, canopy cover predictions, tree density per hectare predictions, and remote sensing variables from optical-infrared, SAR, and topographic sources. The process began with filtering GEDI footprints obtained from the L2A product, covering acquisitions from 2020 to 2022. To ensure high data quality, only footprints with a quality flag of one, indicating reliable signal returns, were retained (Dubayah et al., 2020). Footprints were then spatially filtered using olive grove parcels identified through the Sistema de Información Geográfica de Parcelas Agrícolas (SIGPAC), a cadastral geographic information system that provides high-precision agricultural boundaries (MAPA, 2024). This spatial intersection ensured that only footprints fully within olive groves were included, minimizing contamination from other land uses. Additionally, temporal consistency was maintained by synchronizing GEDI acquisitions with HLS imagery dates to reduce discrepancies between GEDI-LiDAR and Optical-Infrared observations (Masek et al., 2021). To avoid edge effects, particularly those caused by mixed land cover classes, olive grove polygons from the SIGPAC

database were rasterized, and an "erode" filter was applied to reduce parcel boundaries. This process helped eliminate footprints partially overlapping non-olive areas, ensuring data consistency (Indirabai and Nilsson, 2024).

Predictor variables were extracted for each 30m pixel using the Google Earth Engine (GEE) platform, incorporating a comprehensive set of remote sensing features, the resampling method to align all multisource data was Nearest Neighbor. Fully optical-infrared predictors were computed from HLS imagery. To capture additional structural information, SAR texture metrics were derived from ALOS-2 PALSAR-2 imagery using the Gray Level Co-occurrence Matrix (GLCM) method, which provides valuable insights into vegetation patterns through measures of entropy and contrast (Haralick et al., 1973; Shimada et al., 2014). Additionally, topographic variables, including slope, aspect, and elevation, were derived from the SRTM-DEM with a spatial resolution of 30 meters (NASA JPL, 2013). These diverse predictors ensured that the dataset captured the multi-dimensional characteristics necessary for accurate biomass modeling.

To evaluate the influence of waveform noise on GEDI height metrics, we analyzed the relative height (RH) profiles from all valid L2A footprints. For each footprint, we identified the first RH percentile greater than zero, representing the point at which the canopy signal approximately becomes distinguishable from background noise. This analysis was used to assess GEDI's sensitivity across canopy height ranges.

For the validation dataset, the ALS data was used to ensure consistency in both positional and vertical accuracy. The acquisition of this dataset took place in 2020, to ensure alignment with the Optical-Infrared, SAR, and topographic datasets. The validation dataset was processed using the same methodology applied during the biomass modeling phase (Figure 1), enabling a volumetric-based and aboveground biomass density (AGBD) validation approach.

**2.5 Machine Learning Approach for AGBD Prediction**

The selected model for this study was RF, a machine learning algorithm widely recognized for its resilience to noise and effectiveness in addressing overfitting issues. Although RF was initially developed for classification tasks, its use in regression problems, such as biomass estimation, has significantly increased (Pérez-Cutillas et al., 2023). The model was trained using the comprehensive feature dataset detailed in **Table 2**, which integrates optical indices, SAR polarizations and SAR derivatives (PALSAR-2 GLCM metrics), and topographic attributes (SRTM slope, aspect, and elevation). The RF algorithm employs a bootstrap aggregation (bagging) approach, wherein each decision tree is trained on a random subset of the data with replacement. This ensemble method aggregates predictions from multiple trees to reduce variance and improve accuracy (Breiman, 2001).

Model hyperparameters were configured to balance computational efficiency and predictive performance. The maximum tree depth was set to 10, with a minimum of 1 sample per leaf node and 2 samples per split, ensuring that trees remained efficient and avoided overfitting. The number of trees in the forest was fixed at 100, a value determined to provide stable predictions without excessive computational costs. These parameters were selected to optimize model performance within the GEE environment, where complex models with excessive depth or tree counts may become inoperable due to memory constraints. To ensure robust performance across different data configurations, the model was trained under various input combinations of remote sensing variables, as outlined in **Table 2**.

## 2.6 Model Validation and Large-Scale Application

The model validation was conducted through a cross-validation process using the test dataset, based on the internal GEDI L2A–modeled data (Figure 5-a) and independent PNOA-ALS (Figure 5-b). The use of the validation datasets is illustrated in Figure 1, which graphically represents the procedure. The PNOA-ALS point samples (4792 grid samples) constituted the primary dataset for validation (Figure 1), within the task titled *"Data extraction to grid"*. However, cross-validation using the GEDI L2A–modeled data (25% of the total points in Figure 3) allowed for the assessment of model performance against L2A modelled biomass data and provided insights into the most important predictor variables in each evaluated model (Table S1). Standard performance metrics, including the coefficient of determination (R²) and root mean square error (RMSE), were calculated (Eqs. 6-7) as:

$$R^2 = 1 - \frac{\sum_{i=1}^{n}(y_i - \widehat{y_i})^2}{\sum_{i=1}^{n}(y_i - \bar{y})^2} \tag{6}$$

$$\text{RMSE} = \sqrt{\frac{1}{n}\sum_{i=1}^{n}(y_i - \widehat{y_i})^2} \tag{7}$$

Where $n$ is the observed sample size, $y_i$ is the observed value for observation i, $\widehat{y_i}$ is the predicted value, and $\bar{y}$ is the mean of the observed values.

To enhance the robustness of biomass estimation, various input variables derived from optical, radar, and topographic datasets (Table 2) were systematically evaluated. This approach allowed for a comprehensive assessment of feature importance and the individual contribution of each geospatial data source.

The internal model's accuracy was also evaluated against the training dataset derived from the GEDI L2A-based modeling process. For this purpose, the training dataset was randomly split into 75% for model training and 25% for testing (Figure 1). The total number of samples is 44,393, following a filtering process aimed at ensuring that all AGBD ranges are approximately equally represented within the training dataset. This approach was applied to create a balanced training set across the entire AGBD spectrum, as illustrated in the histogram in Figure 3. The performance of the RF model was assessed using the test subset, and the evaluation metrics included the coefficient of determination (R²), root mean square error (RMSE), mean absolute error (MAE), as well as the relative importance of each predictor variable (Table S1).

Initially, optical bands from the HLS dataset, including bands B1 to B9, were tested independently to establish a baseline accuracy. Subsequently, spectral index derivatives were incorporated to assess their impact on biomass prediction accuracy.

**Table 2.Analyses performed on the training set depending on the typology of data sets combination**

| Model | Features | Geospatial dataset |
| --- | --- | --- |
| Optical bands | -B1, B2, B3, B4, B5, B6, B7, B9 | HLS L2A |
| Spectral derivatives | -EVI, GI, GLI, GNDBI, MCARI1, NDWI, NDVI, SAVI | HLS L2A |

| | | |
|---|---|---|
| SAR polarization, textures, and Radar Vegetation Index (RVI) | -HH, HV<br>-HH Contrast, HH Correlation, HH Entropy, HH Inertia<br>-HV Contrast, HV Correlation, HV Entropy, HV Inertia<br>-RVI | ALOS2-PALSAR2 |
| Optical bands, SAR textures, and RVI | -B1, B2, B3, B4, B5, B6, B7, B9<br>-HH Contrast, HH Correlation, HH Entropy, HH Inertia<br>-HV Contrast, HV Correlation, HV Entropy, HV Inertia<br>-RVI | HLS L2A<br>ALOS2-PALSAR2 |
| Optical bands and SAR polarization | -B1, B2, B3, B4, B5, B6, B7<br>-HH, HV | HLS L2A<br>ALOS2-PALSAR2 |
| SAR polarization, textures, RVI, and Topographic data | -HH, HV<br>-HH Contrast, HH Correlation, HH Entropy, HH Inertia<br>-HV Contrast, HV Correlation, HV Entropy, HV Inertia<br>-RVI<br>-Aspect, Elevation, Slope | ALOS2-PALSAR2<br>SRTM v003 |
| Optical bands and Topographic data | -B1, B2, B3, B4, B5, B6, B7, B9<br>-Aspect, Elevation, Slope | HLS L2A<br>SRTM v003 |
| Fully Multi-Source | -B1, B2, B3, B4, B5, B6, B7, B9<br>-EVI, GI, GLI, GNDBI, MCARI1, NDBI, NDVI, SAVI, FVC<br>-HH, HV<br>-HH Contrast, HH Correlation, HH Entropy, HH Inertia<br>-HV Contrast, HV Correlation, HV Entropy, HV Inertia<br>-RVI<br>-Aspect, Elevation, Slope | HLS L2A<br>ALOS2-PALSAR2<br>SRTM v003 |

Note: Acronyms used in this table are as follows: B1-B9 (Spectral bands from Harmonized Landsat-Sentinel), HLS L2A (Harmonized Landsat-Sentinel Level 2A product), EVI (Enhanced Vegetation Index), GI (Green Index), GLI (Green Leaf Index), GNDBI (Green Normalized Difference Built-up Index), MCARI1 (Modified Chlorophyll Absorption Ratio Index 1), NDWI (Normalized Difference Water Index), NDVI (Normalized Difference Vegetation Index), SAVI (Soil-Adjusted Vegetation Index), SAR (Synthetic Aperture Radar), HH (Horizontal-Horizontal polarization), HV (Horizontal-Vertical polarization), Contrast, Correlation, Entropy, and Inertia (GLCM Texture metrics derived from SAR data), RVI (Radar Vegetation Index), ALOS2-PALSAR2 (Advanced Land Observing Satellite-2 Phased Array type L-band Synthetic Aperture Radar-2), SRTM v003 (Shuttle Radar Topography Mission version 3), Aspect, Elevation, and Slope (Topographic variables derived from SRTM).

Further evaluations integrated SAR features derived from ALOS-2 PALSAR-2, considering polarization channels (HH, HV), texture metrics (contrast, correlation, entropy, and inertia), and the RVI. These variables were tested individually and in combination with optical bands to examine their synergy in biomass estimation. To explore topographic influences, SRTM data were added, including terrain-related predictors such as slope, aspect, and elevation. Finally, the most comprehensive

model, termed Fully Multi-Source, incorporated all available data sources, including optical bands, spectral indices, SAR-derived features, and topographic variables.

For large-scale application, the trained model was deployed on the GEE platform, enabling the generation of a high-resolution
AGBD map of olive groves across Europe. Additionally, the model was tested under different regional satellite image coverage scenarios, validating its generalization capability and robustness across diverse agro-climatic conditions. The integration of multi-source remote sensing data within a cloud-based platform ensures scalability and operational feasibility, making it a valuable tool for large-scale biomass assessments.

## 3 Results

**3.1. Training Phase: biomass modelling framework for AGBD inputs and Relative Height Metrics assessment**

The training dataset was compiled using filtered GEDI canopy height data from the study area, primarily covering the extensive olive orchards of Andalusia, and other autonomous communities of Spain. **Figure 3 (a)** illustrates the spatial distribution of the training samples along with the histogram of AGBD values obtained from the volumetric model (GEDI L2A).

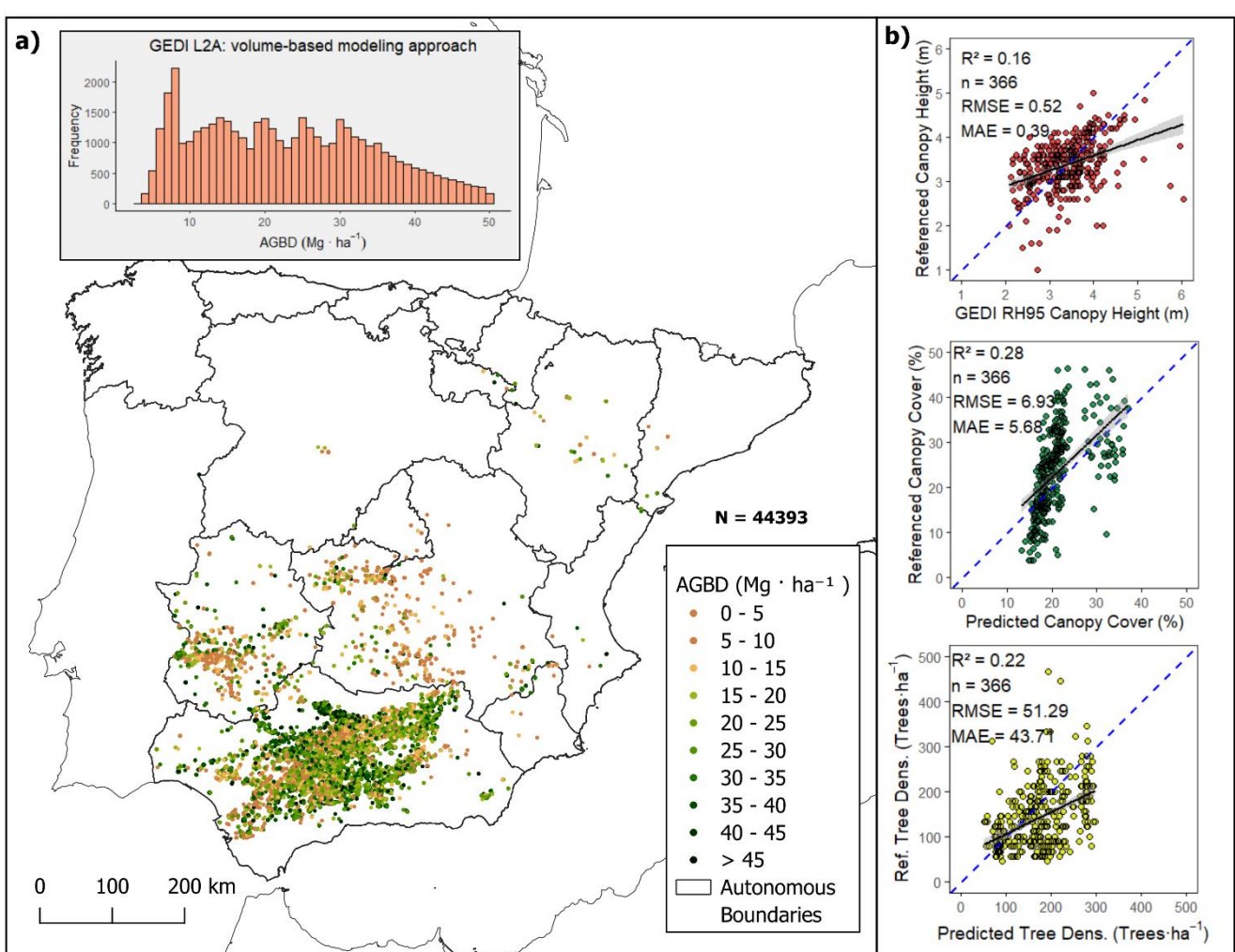

**Figure 3. Spatial distribution of training points (GEDI L2A footprints) filtered for olive orchard plots in Spain from 2020 to 2022. (Figure 3 shows in panel "a" the spatial distribution of the points to which the volumetric model was applied, along with the corresponding histogram of the AGBD results. Panel "b" shows the GEDI vegetation height validation, Total Canopy Cover predictions, and Tree Density predictions)**

The model derived from GEDI L2A enabled the construction of a dataset that spans the entire range of AGBD values for olive

groves, while also maintaining balance across biomass classes. This dataset was used for subsequent training with RF for large-

scale biomass mapping. Figure 3a shows that the frequency distribution is representative across all biomass intervals, which

is critical for the robustness of RF training. The training points are primarily concentrated in southern Spain due to the high

density of olive plantations in the region, which allowed us to capture the variability of cultivation practices across this area.

However, the dataset includes three key variables with a certain level of uncertainty that are essential for estimating biomass

using the proposed approach: (i) vegetation relative height (from GEDI L2A), (ii) canopy cover, and (iii) tree density (trees

per hectare). To validate these variables, a total of 366 manually interpreted reference points were used (Figure 3). For the

GEDI L2A height metric (RH95), validation was performed by selecting pixels with an overlapping GEDI footprint. The same set of reference samples was used for validating canopy cover and tree density estimates.

The canopy height estimates from GEDI L2A were compared to the ALS referenced data using regression analysis, yielding an $R^2$ of 0.16, a mean absolute error (MAE) of 0.39 m, and a root mean square error (RMSE) of 0.52 m. Canopy cover estimates, derived from linear regression trained on PNOA sampling data, were validated against reference values, resulting in an $R^2$ of 0.28, MAE of 5.68%, and RMSE of 6.93%.Finally, tree density predictions obtained through the Random Forest model were also compared to the reference data, with a performance of $R^2 = 0.22$, MAE = 43.71 trees·ha$^{-1}$, and RMSE = 51.29 trees·ha$^{-1}$.

Figure 4a shows the mean Relative Height (RH) waveforms for different vegetation height classes, derived from all available GEDI L2A footprints (n = 44393). The average curves reveal that tall olive trees (>10 m) exhibit progressively increasing RH profiles, clearly separated from the other height classes. In contrast, the waveforms for low and medium vegetation (<5 m) are very similar to each other. For these shorter olive trees, the ground level (height = 0) is reached at approximately RH percentile 50, suggesting that a large portion of the vertical structure signal is lost due to waveform noise. In the high vegetation group (5–10 m), the ground level is reached earlier, around RH percentile 35, while for very tall olives (>10 m) it occurs substantially earlier, around RH percentile 10.

Figure 4b displays the distribution of the first RH percentile greater than zero for each height class, representing the point at which GEDI first detects a return distinguishable from noise. Figure 4b also reflects the overall height composition of the sampled data (Figure 3a), dominated by olive orchards below 5 m in height. The results indicate that most GEDI footprints used for model training correspond to medium-height trees (2.5–5 m), while low and high vegetation classes are less frequent.

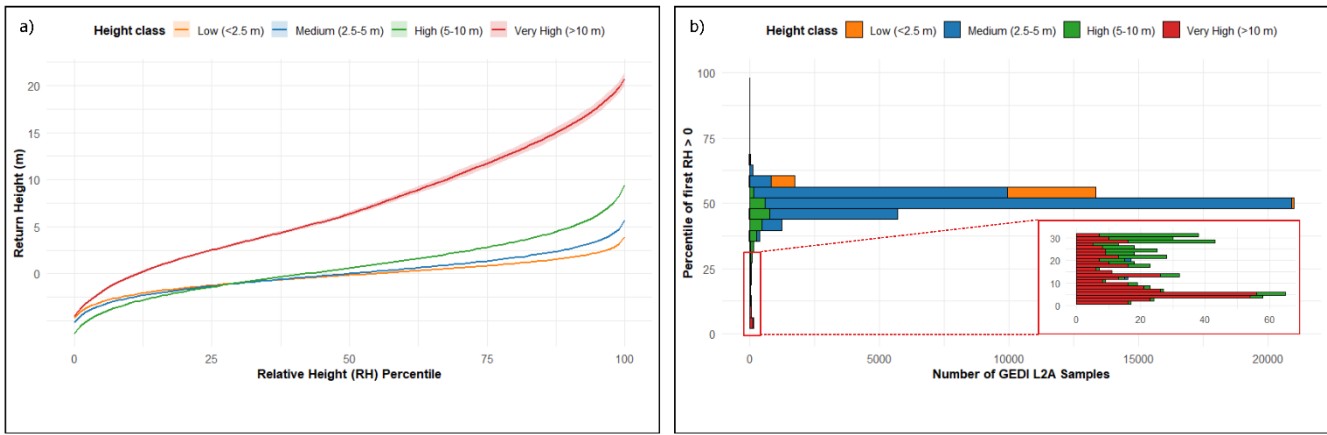

**Figure 4. Relative Height Metrics assessment for training points (n=44393). The panel "a" shows the mean relative height (RH) waveforms for each tree height class, calculated up to RH95. Curves represent the cumulative vertical energy distribution within each GEDI footprint. The panel "b" displays the distribution of the percentile where the first GEDI relative height (RH) value greater than zero was detected, grouped by tree height class.**

### 3.2. Model Performance and AGBD Prediction Accuracy

The predictive performance of the model varied depending on the dataset combinations used during the training phase. The models were evaluated by comparing the RF predictions against the reference dataset, which consisted of 4,782 independent test samples. The "Fully Multi-Source" model consistently achieved one of the highest accuracies, with an $R^2$ of 0.14 and an RMSE of 6.68 Mg·ha$^{-1}$ when compared to the independent ALS-based referenced dataset. When compared to internal L2A-based modeled test dataset, the model yielded an $R^2$ of 0.56 and RMSE of 7.73 Mg·ha$^{-1}$ for GEDI L2A volume-based approach, demonstrating the advantage of integrating multiple remote sensing sources, including optical, SAR, LiDAR, and topographic variables. Among the individual data sources, topographic and optical datasets significantly improved model accuracy (**Figure 5**).

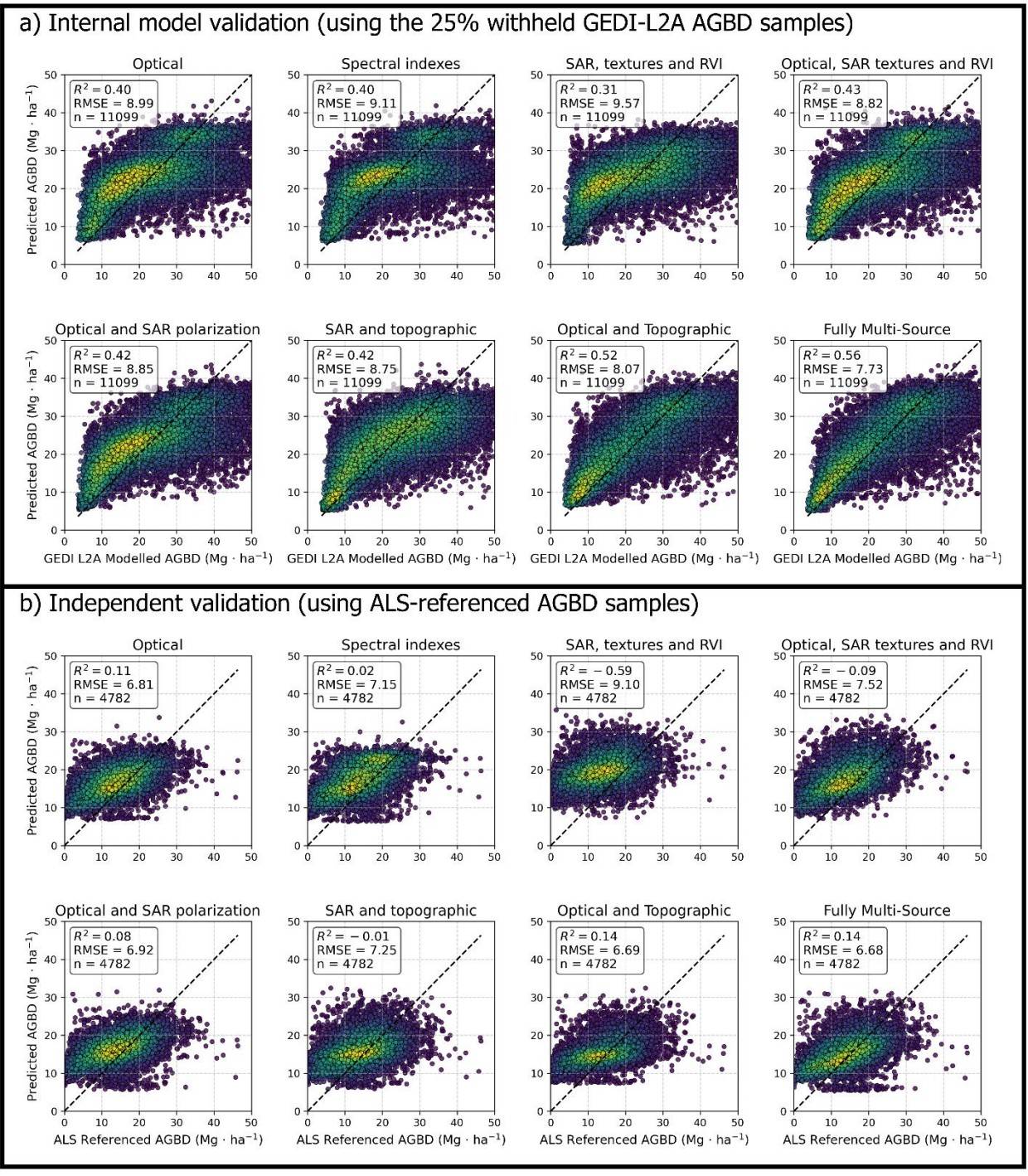

Figure 5. Model validation results using two reference datasets. (a): Internal model validation based on GEDI L2A-derived AGBD samples (25% of the total dataset withheld for testing). (b): Independent validation using high-resolution ALS-referenced AGBD samples. The ALS-based validation represents the most reliable accuracy assessment since it is independent of GEDI-derived

training data and not affected by waveform-related uncertainties. Each panel shows the relationship between predicted and reference AGBD for different predictor combinations (optical, SAR, and topographic). The dashed black line represents the 1:1 line. For interpreting the colour of the dots it is necessary to know that the yellow colour represents high density of dots while the dark blue colour represents low density of dots. The error statistics were computed using the equations 6-7.

The "Optical bands and Topographic data" model achieved the second-best performance, with an $R^2$ of 0.14 and an RMSE of 6.69 Mg·ha$^{-1}$ compared with the independent ALS referenced test dataset and compared with internal L2A modelled dataset with $R^2$ of 0.52 and an RMSE of 8.07. The inclusion of slope and elevation variables proved to be particularly beneficial in cultivated areas with significant terrain variability. Models relying exclusively on optical bands or spectral indices yielded lower predictive power. The "Optical bands" model produced $R^2$=0.11 , while the "Spectral Indexes" model underperformed, with $R^2$=0.02. These results indicated that, while vegetation indices contribute relevant spectral information, they alone are insufficient for accurate biomass estimation. Radar-derived models showed the lowest predictive accuracy.

The "SAR polarization, textures, and RVI" model exhibited the weakest performance, with an $R^2$=-0.59 and RMSE= 9.10 Mg·ha$^{-1}$ compared with the independent ALS referenced dataset (Figure 5). The incorporation of topographic data alongside SAR variables improved the predictions slightly ($R^2$ = -0.01), but RMSE values remained significantly higher compared to models integrating optical data. The combination of optical and SAR data showed moderate improvements over individual sources. The "Optical bands and SAR textures and RVI" model achieved an $R^2$= -0.09, representing a slight improvement over models using only SAR data. Similarly, the "Optical bands and SAR polarization" model reached an $R^2$= 0.08, with corresponding RMSE values of 6.92 Mg·ha$^{-1}$. When comparing the predicted values with the internal GEDI L2A modeled test dataset (as shown in the upper eight scatterplots of Figure 5), the $R^2$ values improve substantially. However, the RMSE values are generally higher, reflecting increased dispersion despite stronger correlation.

Overall, all model configurations demonstrated relatively low accuracy when compared to the independent ALS-based referenced dataset. However, certain combinations proved more effective than others, yielding more reliable results. In particular, the two best-performing configurations were the "Optical and Topographic" and "Fully Multi-Source" models, both achieving similar $R^2$ values and comparable RMSE. These findings highlight that the most accurate models were those that integrated topographic and optical data, with the fully integrated multi-source model producing the best overall performance due to the lowest RMSE. In contrast, models relying exclusively on spectral indices or radar data exhibited lower predictive capacity, reinforcing the importance of integrating multiple remote sensing sources for improved AGBD estimation.

### 3.3. Spatial Distribution and Large-Scale Mapping of AGBD

The spatial distribution of AGBD across Mediterranean olive-growing regions exhibits significant variations at both national and sub-national levels. Spain presents the highest total AGB among the analyzed countries, with extensive areas of high-density biomass predominantly located in Andalusia, a region in southern Spain. The highest biomass values, exceeding 50 Mg·ha$^{-1}$, are concentrated in this southern part of the country, particularly in regions characterized by intensive olive cultivation. Other areas in Spain, such as Castilla-La Mancha and Catalonia, display moderate AGBD values,

ranging between 20 and 40 Mg·ha⁻¹, while lower biomass densities, below 15 Mg·ha⁻¹, are observed in more marginal olive-growing areas (**Figure 6**).

Italy follows Spain in total AGB stock, but the latter is more heterogeneous distribution. For Italy, the highest biomass densities, reaching 40–50 Mg·ha⁻¹, are mainly observed in central and southern Italy, particularly in Tuscany, Apulia, and Calabria. Northern and insular regions such as Sardinia and Sicily exhibit more variable biomass values, with patches of high-density AGBD interspersed with areas below 20 Mg·ha⁻¹. Greece ranks third in total AGBD but displays a distinctive biomass distribution pattern. The highest densities, ranging from 30 to 45 Mg·ha⁻¹, are concentrated in Crete and the Peloponnese,

which are key regions for olive production. Other olive-growing areas in mainland Greece show lower biomass densities, generally below 25 Mg·ha⁻¹, with more fragmented and dispersed high-density patches.

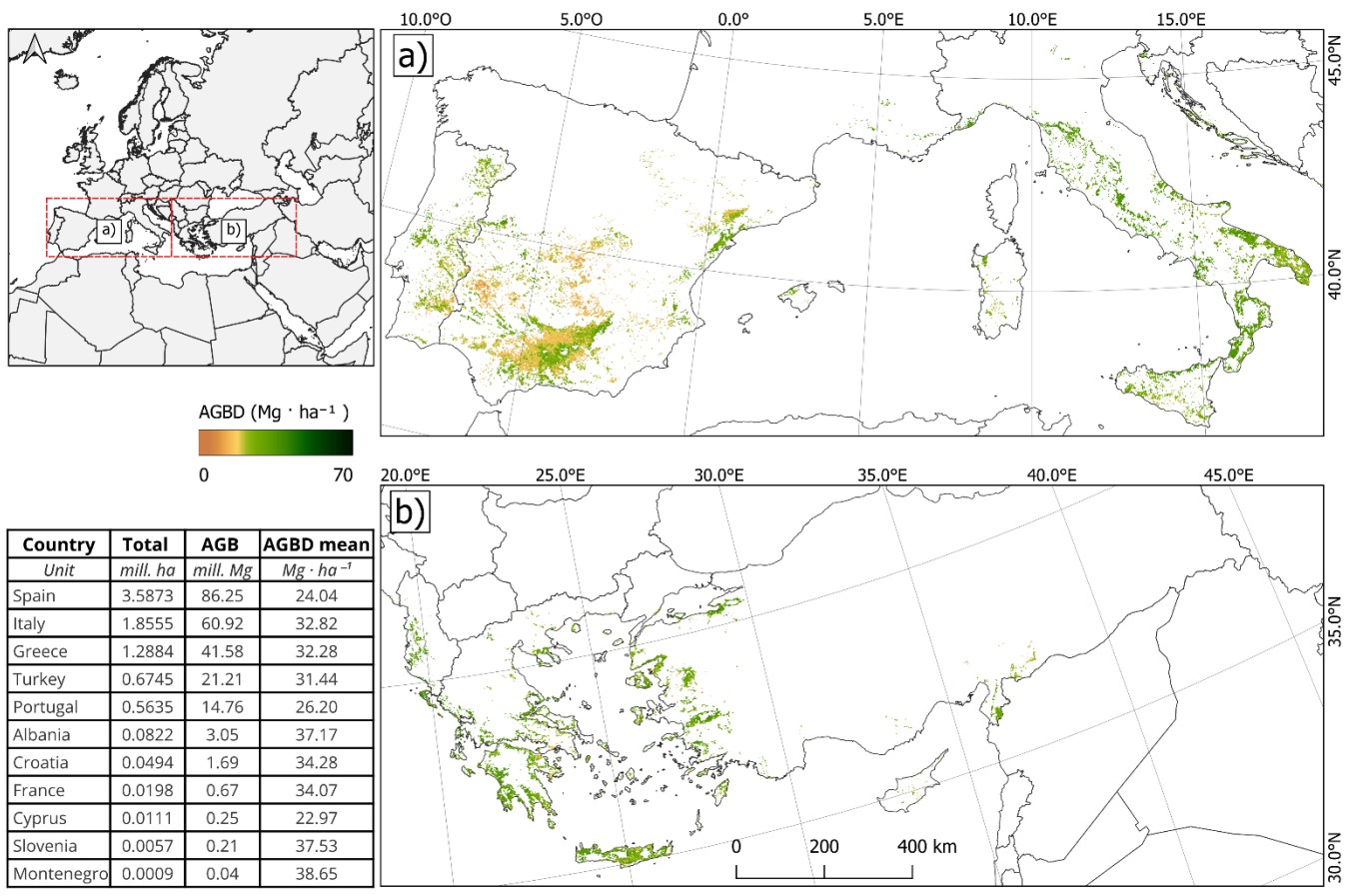

| Country | Total | AGB | AGBD mean |
|---|---|---|---|
| Unit | mill. ha | mill. Mg | Mg · ha⁻¹ |
| Spain | 3.5873 | 86.25 | 24.04 |
| Italy | 1.8555 | 60.92 | 32.82 |
| Greece | 1.2884 | 41.58 | 32.28 |
| Turkey | 0.6745 | 21.21 | 31.44 |
| Portugal | 0.5635 | 14.76 | 26.20 |
| Albania | 0.0822 | 3.05 | 37.17 |
| Croatia | 0.0494 | 1.69 | 34.28 |
| France | 0.0198 | 0.67 | 34.07 |
| Cyprus | 0.0111 | 0.25 | 22.97 |
| Slovenia | 0.0057 | 0.21 | 37.53 |
| Montenegro | 0.0009 | 0.04 | 38.65 |

**Figure 6. The map shows the distribution of AGBD in olive orchards across Mediterranean countries applying the "Fully Multi-Source" model. Panel "a" shows the AGBD distribution in the Iberian Peninsula and Italy, while Panel 'b' displays the AGBD**

**distribution in Greece, Turkey, and surrounding regions. The color gradient represents AGBD values, ranging from 0 (brown) to 70 Mg·ha⁻¹ (dark green). The inset table summarizes the total cultivated area (mill. ha), total AGB stock (mill. Mg), and mean AGBD (Mg·ha⁻¹) for each country.**

The large-scale AGBD mapping provides a detailed representation of biomass gradients across CLC-based olive orchards, enabling a refined analysis at regional and local scales. In all three countries, intensive production zones consistently exhibit higher biomass values, while traditional and less-managed orchards tend to show lower AGBD densities, often below 20 Mg·ha$^{-1}$. These spatial variations emphasize the model's ability to accurately capture biomass heterogeneity across diverse agro-ecological contexts, offering a valuable tool for biomass monitoring and sustainable resource management in Mediterranean olive-growing regions.

### 3.4. Impact of Remote Sensing Data Integration on Model Performance

Each model was analyzed based on the performance of RF in predicting AGBD, using the training dataset, as indicated by the orange block in Figure 1. The dataset was built from previously generated products, all of which carry an associated level of uncertainty that was previously evaluated (Figure 3b). Overall, the results confirm that integrating diverse geospatial datasets enhances the robustness and precision of AGBD predictions. The combination of LiDAR, optical, SAR, and topographic data allowed for a more comprehensive characterization of olive biomass distribution, reducing uncertainty in predictions and improving spatial resolution across Mediterranean agricultural landscapes. Feature importance analysis provided insights into the relative contributions of different predictors to the model. Each model identifies key features relevant to estimation, aiding in the interpretation of their characteristics and influence on predictions.   The integration of spectral indices derived from HLS further enhanced the model, with vegetation index emerging as key variables due to their strong correlation with vegetation vigor and canopy structure provided by GEDI sensor.

Among the spectral indices, MCARI1, Normalized Difference Vegetation Index (NDVI) and Normalized Difference Built-up Index (NDBI) emerged as the most influential predictors of AGBD (Table S1). Optical bands consistently outperformed SAR-derived features, demonstrating greater sensitivity to vegetation structure and biomass variations. Within the SAR datasets, HV polarization contributed the most to the predictions, with simple HV polarization proving more effective than its associated texture metrics. However, models relying solely on SAR features tended to underestimate biomass values, leading to reduced accuracy compared to those integrating optical data.

The results indicate that models integrating spectral data, HV polarization, and topographic variables achieved the highest predictive performance, highlighting the strong influence of topography—particularly slope and elevation—on biomass estimations. Notably, 'Slope' was the second most important predictor in the "Fully Multi-Source" model, accounting for 27% of the total variable importance, second only to NDBI, which contributed 42% (Table S1).In the optical domain, bands B7, B5, and B4 provided the highest predictive value, reinforcing the importance of vegetation-sensitive spectral regions in biomass modeling. Overall, the fully integrated multi-source model provided the most reliable AGBD estimates, reducing RMSE by approximately 20% compared to LiDAR-only models. These findings underscore the necessity of incorporating diverse remote sensing inputs to enhance predictive accuracy and ensure robust biomass assessments in Mediterranean olive orchards.

## 4 Discussion

### 4.1. Model Performance and Data Integration

The volumetric modeling approach for AGBD has proven to be a potentially scalable method. However, as implemented in this study, the modeling phase incorporates multiple sources of error that are subsequently propagated into the final model, representing a primary limitation for the accuracy of this methodology. While the volumetric approach yields realistic biomass estimates at the individual tree level, its application at the geospatial (per-pixel) scale, requiring the derivation of intermediate predictive sub-products, introduces additional uncertainty into the training dataset, which in turn reduces the overall accuracy of the final AGBD estimates.

The validation results of the sub-products, including GEDI L2A canopy height, total canopy cover, and tree density, revealed relatively low predictive performance, which negatively impacted the biomass estimation. The model's heavy dependence on these intermediate variables undermines its reliability, as generating sub-products with limited precision compromises the predictive accuracy of the overall model. Nevertheless, evaluating the performance of each of these components is essential for identifying the model's weaknesses and understanding the key challenges that need to be addressed in future research.

This result aligns with the findings Velázquez-Martí et al. (2014) of who demonstrated that crown diameter and volume are strong predictors of biomass in olive trees. However, the limited availability of large-scale field data remains a significant challenge for validating biomass predictions (Fernández-Sarría et al., 2019). Although volumetric calculations offer a direct estimation approach, uncertainties inherent to dendrometric measurements can propagate into the model, affecting prediction accuracy.

The GEDI RH95 metric revealed several limitations in terms of vertical accuracy ($R^2 = 0.16$, RMSE = 0.52 m) when compared with ALS dataset (Figure 3b). Although the GEDI waveform can effectively monitor biomass in tall vegetation, its performance in olive orchards was characterized by low accuracy. The typical height of olive trees is below the optimal detection threshold of the GEDI waveform, introducing a major source of error because the instrument is unable to capture fine-scale vegetation structure. This finding is consistent with Li et al. (2023), who demonstrated that vegetation shorter than 2.34m is difficult to measure accurately with GEDI, whereas the precision for trees taller than 3 m remains reliable. As a result, GEDI-derived relative height estimates (Figure 3b) often show considerable uncertainty. Therefore, future studies should develop new methodologies to better characterize low-stature vegetation using GEDI or complementary sensors (Chen et al., 2025). Similarly, GEDI struggles to detect tree heights in low-stature canopies such as olive orchards (Atmani et al., 2022). This limitation propagates into the GEDI L2B product and represents the main reason why L2B canopy cover was not used to calculate AGBD in the proposed framework (Figure 1). Consequently, canopy cover had to be estimated independently (Figure 3b) to support biomass modeling, which introduced an additional source of uncertainty.

The Relative Height analysis (Figure 4) further supports this interpretation. Low olive trees (<5 m) exhibit markedly lower waveform amplitudes and flatter RH profiles compared with taller trees. This pattern indicates that a smaller proportion of the GEDI return energy is associated with the canopy, while a greater portion is dominated by system noise. Moreover, the ground

level (height = 0) is typically detected at higher RH percentiles (around RH50) for short vegetation, whereas for trees exceeding 10 m it occurs much earlier (around RH10–30). These results suggest that GEDI's ability to resolve the canopy–ground separation decreases as vegetation height approaches the noise threshold (1–2 m). Similar limitations have been reported in savanna and open-canopy ecosystems, where GEDI height estimates become unreliable below approximately 2–3 m (Hancock et al., 2019; Li et al., 2023). Previous studies also demonstrated that waveform noise and beam sensitivity can delay the detection of the first return, resulting in an artificial compression of the vertical profile in low or sparse canopies (Adam et al., 2020; Wang et al., 2022). Therefore, the reduced amplitude and delayed ground detection observed in low olive canopies likely reflect signal degradation caused by both instrumental noise and limited vertical structure, consistent with the known sensitivity limits of the GEDI sensor.

The model's predictive performance varied depending on the combination of remote sensing datasets used during training. Optical and topographic variables played a crucial role in improving model accuracy, with slope and elevation contributing significantly, especially in regions with complex terrain. This is consistent with previous studies highlighting the influence of topography on vegetation growth and biomass distribution (Indirabai and Nilsson, 2024; Rodríguez-Lizana et al., 2023). Among optical indices, NDBI and NDVI exhibited the highest importance, reflecting their sensitivity to vegetation structure and water content (Kebede and Soromessa, 2018). Notably, spectral bands B7, B5, and B4 contributed valuable information, supporting findings by (Estornell et al., 2012) regarding the relationship between spectral reflectance and biomass estimation. The comparative analysis of different dataset combinations showed that integrating optical, SAR, and topographic variables produced the most accurate predictions, with the fully multi-source model achieving the highest R² and lowest RMSE. This result underscores the benefits of combining complementary data sources to capture the diverse factors influencing biomass. While SAR data, particularly HV polarization, added structural information, its predictive capacity was lower than that of optical and topographic variables. This observation aligns with Shendryk (2022), who noted that SAR-derived metrics are more effective in forest environments due to their sensitivity to canopy structure.

On the other hand, it should be added that the model's scalability and computational efficiency are notable advantages, facilitated by Google Earth Engine's (GEE) cloud-based infrastructure. GEE's capacity to process large geospatial datasets enables the application of the model over extensive areas without significant computational constraints. This scalability is essential for generating high-resolution biomass maps that support carbon stock assessments and sustainable land management (Figure S1). Furthermore, the use of Random Forest regression within GEE enhances model robustness (Pérez-Cutillas et al., 2023), due to its resilience to noise and non-linear relationships. Overall, the integration of diverse remote sensing datasets, combined with the computational capabilities of GEE, enables accurate and scalable AGBD estimation across Mediterranean olive orchards. This methodological approach provides a practical solution for large-scale biomass monitoring, aligning with current efforts to improve carbon accounting and resource management in agricultural landscapes.

Several studies have employed GEDI datasets or similar remote sensing technologies, such as ICESAT, to map biomass, primarily focusing on extensive forested regions (Indirabai and Nilsson, 2024; Jiang et al., 2022; Li et al., 2024; López-Serrano et al., 2019; May et al., 2024; Shendryk, 2022). However, the application of remote sensing techniques to perennial crops,

particularly olive orchards, remains relatively unexplored. While a few studies have quantified biomass in olive trees and assessed pruning residues (Estornell et al., 2015; Rodríguez-Lizana et al., 2023; Velázquez-Martí et al., 2014), these efforts have generally been limited to localized contexts, highlighting the need for scalable approaches capable of providing large-scale estimates.

## 4.2. Implications for Biomass Estimation in Mediterranean Olive Orchards

The spatial mapping of biomass in olive groves provides practical benefits for farmers, environmental managers, and policymakers. The biomass quantification supports carbon accounting, enabling more sustainable agricultural practices through optimized residue management and reduced carbon emissions. Identifying areas with high biomass density facilitates decisions on biomass plant locations, promoting efficient resource utilization and enhancing the economic viability of energy production from agricultural residues (Rodríguez-Lizana et al., 2023; Velázquez-Martí et al., 2014).

By quantifying biomass accumulation, the model provides a means to explore the estimation of carbon stocks and potential trends in carbon sequestration, contributing to the understanding of their role in global climate mitigation efforts, while recognizing that these estimates are subject to. High-density olive orchards, with AGBD values reaching 50 Mg·ha$^{-1}$, may act as significant carbon sinks, potentially offsetting atmospheric $CO_2$, though precise quantification at fine scales remains limited. Identifying these areas promotes sustainable land management practices, including the use of pruning residues as organic soil amendments, which further enhances soil carbon retention while reducing emissions from conventional waste disposal (Kebede and Soromessa, 2018).

Moreover, AGBD predictions help improve crop efficiency by identifying orchards with lower biomass productivity, enabling targeted interventions such as optimized irrigation, nutrient management, and canopy pruning to enhance growth and yield. At the regional level, the model supports biomass inventories, providing essential data for both policymakers and farmers to assess the feasibility of biomass-based energy production. For example, the geostatistical approach of Rodríguez-Lizana et al. (2023) demonstrates the potential to identify optimal biomass plant locations, reducing transportation costs and emissions while supporting renewable energy within local economies. Additionally, monitoring biomass dynamics over time allows for evaluating the long-term sustainability of agricultural practices, ensuring that productivity improvements do not compromise soil health or carbon sequestration capacity. By integrating remote sensing technologies like GEDI and multispectral imagery, the model offers a scalable and cost-effective solution to support climate-smart agriculture and sustainable land use in Mediterranean olive orchards.

The biomass values obtained in this study are consistent with prior research estimating biomass in both forested and agricultural landscapes. Unlike dense and unordered forest vegetation, olive groves exhibit structured planting frameworks and lower canopy heights, resulting in lower biomass values per unit area (Fernández-Sarría et al., 2019). Regarding model performance, previous studies have reported similar predictive accuracy, despite variations in vegetation types and geographic settings. For

instance, Li et al. (2024) achieved an R² of 0.59, while in this study the validation against independent ALS data yielded a
635 lower R² of 0.14, reflecting the inherent limitations of GEDI-derived metrics when applied to low-stature vegetation.

Despite these constraints, the proposed framework provides an exploratory but scalable approach for assessing AGBD across Mediterranean olive landscapes. Remote sensing models calibrated with satellite data may struggle to capture fine-scale variability within orchards, particularly where tree height approaches the GEDI noise threshold, limiting precision at local scales. While high resolution LiDAR approaches have demonstrated high precision in small-scale biomass assessments
(Estornell et al., 2012; Fernández-Sarría et al., 2019), their limited spatial coverage restricts their applicability for regional or continental-scale assessments. Consequently, integrating high-resolution UAV data with large-scale satellite observations could enhance model performance, enabling more accurate and scalable biomass mapping in Mediterranean olive orchards.

### 4.3. Limitations of AGBD model and Future Research

The AGBD model developed in this study presents several limitations related to input data variability, environmental factors, computational constraints, and generalization potential. A key challenge is the variability and accuracy of input data, particularly the dependence on volumetric estimates derived from dendrometric measurements. These measurements, while essential, introduce uncertainties due to variations in tree structure and orchard management practices (Rodríguez-Lizana et al., 2023; Velázquez-Martí et al., 2014). Additionally, the spatial filtering of GEDI footprints to isolate olive groves may
exclude mixed land covers, further affecting data accuracy; however, spatial misalignments between the various remote sensing layers and the GEDI footprints prevent the model from capturing finer-scale details within each pixel. One potential limitation of the model is that the estimated AGBD mean (**Figure 6**) values for Spain are relatively low compared to other countries within the study area, despite Spain having the largest olive-growing area in the European Union. This discrepancy raises the question of whether olive trees in Spain inherently have lower biomass density or if the model overestimates biomass
density in regions with lower olive tree density. A key factor influencing this outcome is the environmental conditions prevalent in Spain, particularly the arid and semi-arid climates that characterize extensive areas of the country. The primary predictors used in the model (NDBI, NDVI, and MCARI1) are sensitive to vegetation cover and moisture content (Fern et al., 2018), which may result in lower predicted AGBD values in regions with sparse vegetation and greater exposure of bare soil. Conversely, areas with more abundant vegetation cover, such as Slovenia, Croatia, and Italy, exhibit higher average AGBD
values, highlighting the influence of spectral indices in capturing vegetation density and health.

This effect underscores the importance of considering environmental factors, such as precipitation and water availability, directly impacting biomass accumulation in olive orchards, but these factors were not directly integrated into the model. The observed influence of slope on model performance in Spain suggests that including climatic and soil properties could enhance prediction accuracy, as reported in previous studies on olive groves and other crops (Kebede and Soromessa, 2018; Rodríguez-
665 Lizana et al., 2023). At finer spatial scales, biomass and residue production have been shown to vary significantly depending on canopy cover (Rodríguez-Lizana et al., 2023). This variability may also be associated with other factors influencing

biomass, such as differences in irrigation practices and soil composition, particularly soil organic carbon (SOC) and clay content, can also impact biomass estimates, highlighting the need for more comprehensive datasets in future research. Consequently, while Spain's extensive olive cultivation contributes significantly to total biomass production, the average biomass density is comparatively lower possibly due to the region's climatic constraints.

From a computational perspective, the cloud-based GEE platform facilitates large-scale biomass mapping, yet its reliance on pre-processed datasets can limit model customization and input diversity. The RF algorithm, while robust and efficient, is sensitive to data quality and may exhibit reduced performance when input data are sparse or noisy (Shendryk, 2022). Moreover, the computational complexity of integrating multisource data, including optical, SAR, and LiDAR metrics, requires optimizing model parameters to balance accuracy and processing time, as demonstrated in large-scale biomass studies using GEDI and Sentinel data (Indirabai and Nilsson, 2024; Shendryk, 2022) .

Remote sensing limitations also affect biomass estimation, particularly in regions with heterogeneous canopy structures. SAR data, despite its ability to penetrate cloud cover, exhibited limited predictive capacity compared to optical data, consistent with findings in previous studies (Kellner et al., 2023; Rodríguez-Lizana et al., 2023). The saturation effect observed in high-biomass areas remains a challenge, although combining SAR and optical data partially mitigated this issue. Additionally, the spatial resolution of GEDI footprints (25 m) may not capture fine-scale variations within olive orchards, especially in areas with irregular planting patterns. Additionally, ALOS PALSAR has proven to be a reliable predictor of biomass in complex forest structures. However, in highly structured plantation frameworks with relatively low canopies, its performance is suboptimal. Therefore, the use of Sentinel-1 (C-band) data may offer a promising alternative for future approaches, given it.

Uncertainty in biomass estimations arises from both input data variability and model assumptions. The volumetric approach, while scalable, inherently carries higher uncertainty than allometric methods, which rely on direct measurements of tree height and diameter at tree breast. The exclusion of certain environmental variables, such as soil moisture and nutrient availability, further contributes to prediction uncertainty. Nevertheless, the most critical factor for achieving accurate results is the availability of a robust and reliable training dataset. In this regard, GEDI has not proven to be a dependable source for estimating tree height in olive groves. Furthermore, the associated subproducts "canopy cover" and "tree density" also exhibited limited reliability, which ultimately constrained the model's performance and increased prediction uncertainty. The model's performance, with $R^2$ of 0.14, indicates a limited predictive capability. This reduced accuracy is primarily attributed to the generalized nature of the training dataset and the low reliability of GEDI-derived subproducts, which are less effective for low-stature vegetation. These limitations are clearly reflected in the poor agreement observed when comparing model estimates with the ALS-referenced validation dataset.

Model generalization and transferability represent additional challenges. While the model demonstrated reasonable accuracy in Spanish olive groves, its applicability to other regions may be limited by differences in climate, soil properties, and orchard management practices. However, the relatively consistent environmental conditions across major olive-growing regions increase the likelihood of successful model transfer, provided that region-specific calibration is performed (Rodríguez-Lizana et al., 2023). Future research should focus on expanding the training dataset to include diverse environmental conditions and

incorporating additional variables such as soil texture, SOC, and irrigation regimes to enhance model robustness and generalizability.

The integration of optical and radar data was also considered as a potential way to mitigate the uncertainty observed in low-stature vegetation; however, the current modelling framework imposes certain constraints. Because the AGBD estimation approach was trained using a dataset inherently biased toward biomass values derived from GEDI RH95 metrics and related subproducts, the model shows limited flexibility to capture the broader variability in biomass that could be explained by optical and SAR variables. This intrinsic bias constrains the predictive potential of these additional data sources, thereby reducing their overall contribution within the current framework configuration

Moreover, to improve the accuracy of biomass estimation models in olive groves, it is essential to avoid using subproducts derived from intermediate estimations, as they introduce considerable uncertainty that ultimately degrades model performance. This study demonstrates that training models with the most robust and direct input data is a critical factor, particularly given the poor reliability observed in canopy height, canopy cover, and tree density subproducts derived from GEDI volume-based biomass modelling approach. For future research, the use of artificial intelligence techniques applied to high-resolution optical imagery, such as that provided by the Spanish PNOA program, may offer an effective solution for tree segmentation, thereby eliminating the need for GEDI-based height estimates and avoiding the introduction of uncertainty from derived products. This would result in a more reliable training dataset and significantly improve model accuracy. In addition, integrating alternative remote sensing sources such as Sentinel-1 and Sentinel-2 (Adrah et al., 2025) could enhance the spatial resolution of the final product and better capture fine-scale structural variations for agricultural monitoring.

## 5 Conclusion

This study explored the feasibility of estimating aboveground biomass in large-scale olive plantations by integrating multi-source remote sensing data. Although the model does not yet meet the resolution or precision required for operational crop-level monitoring, it offers a valuable proof-of-concept for the combined use of GEDI and ancillary remote sensing data in structured orchards.

The integration of GEDI LiDAR-derived metrics with optical, SAR, and topographic variables, yielded moderate predictive performance, particularly when evaluated against an independent ALS reference dataset. This result highlights limitations in model generalization at fine spatial scales, although the fully multi-source configuration achieved the highest relative accuracy. Among the predictors, NDBI, slope, and MCARI1 were the most influential, underscoring the importance of both vegetation structure and environmental conditions in biomass modelling.

The volumetric approach based on GEDI L2A height metrics, combined with estimated canopy cover and tree density was constrained by the limited accuracy of these subproducts when used as training inputs. While the method showed reasonable precision at the individual tree level, applying it to grid-based modelling substantially increased uncertainty. These weaknesses

limit its applicability for high-resolution crop monitoring or multi-temporal analysis, though the RMSE values obtained remain acceptable for regional-scale biomass assessment.

The model's large-scale implementation via GEE supports rapid biomass mapping over extensive areas, enabling national-level carbon stock estimation and resource management. Its scalability and automation potential make it suitable for institutional applications and could also provide decision support tools for farmers aiming to optimise irrigation, nutrient input, and biomass use.

Future work should focus on improving training data quality through artificial intelligence approaches and incorporating
additional variables such as climate, soil properties, and irrigation practices. Expanding the training dataset across a broader range of agroecological contexts would enhance the model's robustness and transferability.

Rather than providing an operational solution, this study underscores the challenges and opportunities of using spaceborne LiDAR in low-stature perennial systems, laying the groundwork for future refinements in large-scale biomass estimation.

**Code availability**

Code is available upon request from the corresponding author.

**Author contributions.**

FC: methodology, data processing, visualization, results analysis, and writing. MC: conceptualization, work coordination, paper review, and editing. MASM: conceptualization, work coordination, paper review, and editing. PPC: conceptualization, methodology, results analysis, writing.

**Acknowledgements**

We are grateful to Andrew Feldman, who served as associate editor, and anonymous reviewers for their careful and constructive reviews, which greatly contributed to improving the quality and clarity of our work. This work was supported by two primary sources: in part by the project PID2021-128896OB-I00 from the Spanish Ministry of Science, Innovation and Universities (co-funded by EU FEDER), and in part by the project OLIVCHAR (Ref: TED2021-131907B-I00), financed by
755 MCIN/AEI/10.13039/501100011033 and the European NextGenerationEU/PRTR. We acknowledge support of the publication fee by the CSIC Open Access Publication Support Initiative through its Unit of Information Resources for Research (URICI).

**Competing interests**

The contact author has declared that none of the authors has any competing interests.

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
