# Peer review of "Multi-Source Remote Sensing for large-scale biomass estimation in mediterranean olive orchards using GEDI LiDAR and Machine Learning"

_EGUsphere, 2025_

## Author Response (AR1)

The authors sincerely appreciate the referees' suggestions and comments, which have significantly contributed to improving the quality and clarity of the present manuscript. All suggested revisions have been incorporated into the manuscript, and the responses to the reviewers are highlighted below in blue.

We sincerely appreciate the opportunity to revise the manuscript and remain open to any further modifications that the reviewers or the editor may consider necessary.

**Referee 1:**

This paper proposes a method for estimating aboveground biomass density (AGBD) in olive orchards by combining GEDI L2A height data with orthophoto-derived canopy cover through a volumetric approach. Specifically, crown volume is defined as the overlap between the GEDI L2A footprint and canopy cover, and this volume is then converted to AGBD based on a series of assumptions regarding tree density, stem-to-canopy height ratio, and wood density. Finally, the GEDI L2A-derived AGBD is further predicted using satellite imagery and compared against the GEDI L4A AGBD product as a form of "validation". I think, the use of a volumetric method for AGBD estimation is relevant, particularly in plantation settings such as olive orchards, where tree density, stem-to-canopy height ratio, and wood density can be reliably estimated or sourced from the literature. However, there are three major concerns with how the method is applied in this study:

- First, the authors did not validate their product against field measurements or a more robust method for AGBD estimation (e.g., airborne LiDAR combined with allometric equations). Comparing results only against the GEDI L4A AGBD product is insufficient, as both products could share similar biases or errors.

  We acknowledge the reviewer's concern that comparing the GEDI L4A product with the derived GEDI L2A product may be insufficient for validation. To address this, we have replaced the validation approach in the revised manuscript.

  To strengthen the validity of our volumetric AGBD estimations, we integrated a high-resolution reference dataset from PNOA (Plan Nacional de Ortofotografía Aérea). Specifically, the Canopy Digital Surface Model (DSM) was used to derive tree crown height, while crown diameters were extracted

from orthophotographs. These metrics were processed analogously to the GEDI L2A-based workflow, providing a consistent volumetric estimation framework. This allowed a cross-validation of the GEDI-based AGBD predictions with detailed, spatially precise structural metrics. Additionally, tree crown diameters were measured using PNOA orthophotography.

We argue that this method offers a robust validation for vegetation heights derived from GEDI L2A, as well as the GPR and linear models that estimate tree density and canopy cover.

Furthermore, the volumetric (hemispherical) model applied to the PNOA high-resolution data is processed analogously to the GEDI L2A volumetric approach. To validate the AGBD estimates derived from GEDI L2A footprints, we compared them with AGBD estimations from PNOA-derived crown metrics. These comparisons used the same hemispherical crown volume model, enabling consistency in methodology. The incorporation of the crown occupation factor (Velázquez-Martí et al., 2014) and fixed wood density values ensured compatibility between both datasets. Given the high spatial and vertical accuracy of PNOA-ALS data, this approach serves as an effective substitute for ground-truth data in large-scale biomass modelling. These improvements have been described and developed in section 2.4

Second, the use of a volumetric approach at the footprint level based on GEDI L2A data is questionable. The L2A product has a ~25 m footprint, a positional accuracy of ~10 m, and a vertical accuracy of ~5 m. Combining this with orthophoto-derived canopy cover to estimate biomass is ambitious but fundamentally flawed, as it risks substantial spatial mismatches. Given the number of assumptions involved and the lack of field validation, it is difficult to assess the reliability of the resulting AGBD estimates.

We acknowledge that the GEDI instrument may not be the optimal tool for achieving the highest precision in biomass estimation. However, due to the

limited availability of reliable field validation data, we have incorporated PNOA-based validation in the revised manuscript to strengthen our analysis.

The ALS-derived AGBD estimates, computed using the same volumetric model, were used to assess the reliability of GEDI-based predictions. Results showed consistent spatial patterns, although discrepancies were observed mainly in areas with large areas of bare soil, where canopy detection from satellite images is more uncertain. However, this issue was partially mitigated by the use of ground-adjusted spectral indices (such as SAVI), which improved the final outcomes. This external validation step provided critical support for evaluating the robustness of the model in the absence of direct field biomass measurements.

The primary objective of this study is to evaluate GEDI's performance in generating a comprehensive dataset, leveraging its key advantage: unprecedented spatial coverage across diverse ecosystems. While absolute precision may be constrained by sensor limitations, GEDI's global sampling capability provides unique value for large-scale biomass mapping, a critical gap in current remote sensing approaches.

These points have been described and highlighted in sections 2.4 and 2.6 of the methodology. Furthermore, they have been expanded upon in the "Results" section (sub-sections 3.1 and 3.2).

- Third, it is unclear why the GEDI L2A-derived AGBD is subsequently predicted using satellite imagery at the footprint level. If the goal is to generate wall-to-wall maps, the issue of spatial mismatch remains. Moreover, the manuscript currently reads as if the L2A product itself was used as a predictor for estimating the GEDI L2A-derived AGBD, which, if true, would introduce severe data leakage and invalidate the approach.

We appreciate the opportunity to clarify the methodological framework, which may not have been sufficiently articulated in the original manuscript. Our approach employs a volumetric method at a 30-meter resolution to estimate tree volume, using the AGBD derived from GEDI L2A as the target variable and the predictors listed in Table 2 as input features. This strategy is necessary due to

the fundamental differences between the discrete footprint sampling of GEDI and the continuous spatial coverage offered by conventional remote sensing imagery.

While we recognize that using simulated GEDI L2A-derived AGBD data for model training presents certain limitations, we emphasize that evaluating this specific methodology constitutes a fundamental objective of our research. Beyond biomass estimation, our study specifically examines Random Forest's capacity to predict AGBD with particular application to olive grove ecosystems.

To address potential concerns about methodological validity, we have incorporated high-resolution PNOA validation in the revised manuscript. This independent dataset provides crucial insights into the accuracy of our simulated GEDI-derived inputs during the AGBD modeling phase. We maintain that our approach represents a valid and innovative strategy for overcoming the inherent challenges of working with GEDI's discontinuous sampling pattern while leveraging multi-source remote sensing data synergies.

Unfortunately, the paper also reads as an incoherent combination of machine learning approaches and derived results that are neither sufficiently explained nor properly validated. For example:

The tree density prediction using Gaussian Process Regression lacks a clear explanation of the training data, validation process, or accuracy assessment.

The Gaussian Process Regression (GPR) model was implemented in the first version of the manuscript to estimate tree density (expressed as trees per hectare) through the prediction of cultivation framework patterns. However, we replaced GPR by Random Forest to estimate tree density. This methodological step was essential for assigning accurate labeled tree density values to our dataset. For model development, we utilized a training dataset comprising 849 labeled GEDI footprints representing distinct cultivation frameworks (including both regular and wide spacing configurations. The performance metrics and detailed results from the RF approach are comprehensively presented in the revised manuscript version in lines 407-408.

The canopy cover prediction from aerial imagery using linear regression appears to be based on 250 points derived from orthophotos, but there is insufficient methodological detail or accuracy evaluation.

The canopy cover estimation model was developed using linear regression with the Soil Adjusted Vegetation Index (SAVI) as the primary predictor. To enhance model sensitivity to varying ground conditions, we implemented a dynamic adjustment of the SAVI's "L" factor based on the Bare Soil Index (BSI). This adaptive approach significantly improves model performance in olive orchard environments where soil exposure varies considerably. It is explained in section 2.4.1 (lines 241-248).

For ground truth data, we manually annotated 263 reference points using high-resolution PNOA orthophotography to obtain precise canopy cover measurements. These carefully selected validation points represent the full range of canopy conditions observed in our study area. The model's performance metrics, including accuracy assessments and validation results, are thoroughly documented in the revised manuscript.

The AGBD estimation via the volumetric method seems to depend on unvalidated tree density and canopy cover products.

The accuracy assessment results for the canopy cover model and tree density are presented in detail in the revised manuscript (section 3.1, lines 397-408, and Figure 3). We acknowledge that one of the main weaknesses of the proposed approach lies in its reliance on intermediate products that introduce significant uncertainty into the final outcome. This limitation has been thoroughly analyzed and discussed in the Discussion section of the revised manuscript.

The prediction of AGBD estimates from the volumetric method using Random Forest applied to satellite imagery lacks a clear justification, especially since a direct comparison between the GEDI L2A-derived and L4A AGBD products would have been more straightforward.

We thank the reviewer for raising a valid point - a direct comparison between our modeled "GEDI L2A-derived AGBD" and the official GEDI L4A product would indeed provide a more straightforward validation. However, as previously noted, this study

focuses on two distinct methodological approaches: (1) volumetric and AGBD modeling, and (2) large-scale mapping using modeled/simulated data. This dual focus is essential because a primary objective is assessing biomass resource availability in European olive groves, which requires continuous wall-to-wall mapping that can only be achieved by using modeled data as predictors in our training process.

Regarding potential data leakage concerns, we confirm that we have not used GEDI L4A AGBD data to inform our GEDI L2A-derived AGBD estimations. As clearly outlined in Table 2 of our manuscript, we have meticulously documented all predictors used for each model configuration to maintain complete transparency in our methodology. This separation ensures the integrity of our modeling approach while addressing the study's core goal of generating continuous biomass maps for olive cultivation systems.

Finally, the paper also lacks a clear flow, and many concepts are referenced as common knowledge without proper introduction or explanation. There's a lot of guesswork involved, as key details seem to be assumed rather than explained. It appears that there is a misunderstanding or lack of clarity regarding how the L4A product was derived.

We acknowledge that certain methodological processes in the initial manuscript version lacked sufficient clarity. We appreciate this observation and confirm that these aspects have been thoroughly refined and explicitly detailed in the revised version of the manuscript to ensure complete transparency and reproducibility of our approach.

Additionally, the resolution at which you are working is unclear. I didn't proceed with the discussion section, as my concerns haven't been addressed earlier in the paper, making it difficult to engage with that part meaningfully.

All datasets were standardized to a 30-meter spatial resolution using nearest-neighbor resampling to maintain data integrity during the harmonization process. We acknowledge that this critical preprocessing step was not sufficiently detailed in the original manuscript. To address this, we include a comprehensive schematic diagram (Figure 1) illustrating the complete data preprocessing workflow.

In short, while the study introduces interesting ideas, the lack of rigorous validation and the reliance on stacked, unverified models undermine confidence in the results.

We have implemented a more rigorous validation protocol for our AGBD modeling and Random Forest predictions, including enhanced statistical measures and independent accuracy assessment. These methodological improvements are fully documented in the revised manuscript

Specific comments:

P1, L15: The statement "provided more accurate predictions than the GEDI L4A product" is misleading without field data for validation.

This statement was modified in the new manuscript version:

"The volumetric approach, based on GEDI L2A canopy height and dendrometric parameters, offered a promising framework for large-scale biomass estimation using lidar technologies on satellite platforms"

P1, L19: Is olive tree biomass correlated with olive production? It would be useful to clarify this relationship.

The spatial distribution of predicted biomass highlights the Mediterranean countries with the highest total biomass stocks, particularly Spain, Italy, and Greece. This geographic pattern strongly correlates with olive oil production statistics, reflecting the dominant coverage of olive groves in these regions' agricultural landscapes. But in this context we only intend to indicate the results of the large-scale mapping by highlighting the most important countries (lines 18-19).

P2, L1: The background on remote sensing methods for biomass estimation seems insufficient. Could you expand on the approaches typically used?

Our review of remote sensing techniques in the manuscript includes the most relevant and currently established methods for biomass estimation: ALS, RPAS, optical imagery, and SAR. We have intentionally limited this section to maintain focus on approaches directly applicable to our study, while keeping the introduction section appropriately concise.

P2, L40-41: To my knowledge, SAR, especially L-band, can measure standing dead trees. Do you have a reference to support this?

We acknowledge this observation and agree that the original wording could lead to misinterpretations. The text has been carefully revised to:

"Nonetheless, SAR can also detect dead biomass, especially when structural elements persist". (lines 44-45)

P2, L46-47: Most natural forests are more structurally complex than a plantation. This statement contradicts your assumptions about tree density and structure. Could you clarify this?

This paragraph was rewritten:

"Forest ecosystems typically exhibit complex structures shaped by natural processes, while olive orchards are generally more structurally ordered due to intentional planting arrangements. However, this apparent regularity can be misleading, as olive plantations still display substantial structural variability across sites. Differences in planting frameworks, irrigation regimes, pruning practices, and management intensities introduce considerable heterogeneity, even within this controlled agricultural context". (lines 49-53)

P2, L57-58 and P3, L63-64: There are existing allometric databases for trees, including olive trees (e.g. Tallo). Have you considered these?

Yes, they were considered; these databases are very helpful, especially for the forestry sector. There are a wide variety of possibilities in this field; however, for agriculture, they are flawed due to the limited availability of samples and the low accuracy of sample locations. We have included the presence of this kind of database in the manuscript:

"Conventionally, forest inventories have been the primary source of biomass data for national assessments (Hunka et al., 2025; Nesha et al., 2022), where the availability of allometric databases from diverse locations supports their implementation (Jucker et al., 2022). These methods are widely recognized for their accuracy and are extensively applied in forestry sector inventories" (lines 60-64)

P3, L64-66: Isn't it the opposite? If you know the volume of each tree, there should be less uncertainty, assuming wood density remains constant. Could you clarify?

Yes, assuming you know the volume of each tree the uncertainty of trees should be less, but, even so the GEDI instrument precision doesn't' allow a high accuracy to extract metric heights. We have rewritten this section to avoid misunderstandings:

"Conversely, volumetric models represent a scalable and potentially accurate alternative for biomass estimation. However, when applied to heterogeneous landscapes, their reliability strongly depends on the resolution and accuracy of the input data. In the case of GEDI, while it provides valuable structural information, limitations in both vertical and horizontal precision introduce considerable uncertainty". (lines 71-74)

P3, L83: I'm still unclear on the main challenges of biomass estimation. Is it data scarcity, scale, or something else?

The purpose of this paragraph is to highlight the paucity of studies that use remote sensing to estimate biomass in the form of maps. This has been rewritten:

"Given the limited availability of remote sensing-based biomass estimates in olive groves". (line 90)

P3, L88: How exactly do you ensure spatial consistency? Given the misalignment between GEDI, satellite imagery, and orthophotos, this is a critical issue to address.

In this context, "spatial consistency" is not referred as the method to align the multiple layers. It refers to the correct application of the generated model to the correct land use (olive orchards). This has been rewritten in line 96.

P3, L89: The jump to carbon sequestration seems sudden. Could you link this to biomass first? Are olive orchards typically used for carbon sequestration? This feels somewhat out of context.

Yes, it has been rewritten:

"Ultimately, this research aims to provide a tool for monitoring carbon stocks in form of biomass in olive orchards. In the case of olive orchards, this is particularly relevant given the possibility of transforming residual biomass into stable carbon forms, enabling the development of carbon sequestration protocols". (lines 96-99)

P4, L93: The first sentence should be deleted, as it doesn't seem relevant or necessary.

Yes, this sentence is not necessary.

P4, L105: The proper notation should be Mg/ha or t/ha, not Ton/ha. Please apply this notation consistently throughout the text.

Yes, we agree we need to apply this notation throughout the text, we have changed it in the new manuscript.

P5, L08: I'm confused about the contents of your training and validation dataset. Are you using GEDI, SAR, and optical data? This hasn't been clearly mentioned. Or do you mean that your coverage provides a robust and variable area for training and validating your model?

The geographical extent of olive orchards across Spain provides a robust dataset. This wide coverage ensures variability in environmental and structural conditions, enhancing the model's reliability.

We believe it is clear in this context as it precedes a sentence indicating the large coverage of olive groves in Spain. (lines 110-111)

P5, L113-118: This seems more suited for the introduction rather than the methods section. Could you move it accordingly?

We appreciate this suggestion. While this point could indeed fit within the introduction, we believe the 'Study Area' section is also appropriate as it directly explains the significance of Spain's olive orchard coverage in the context of our methodology. This helps justify the representativeness of our dataset for selecting footprints within target areas.

P5, L120: The motivation for not using the L4A product seems to be missing in the introduction. Could you elaborate on why it was not used instead of developing a new product?

The motivation for not using L4A is expressed in the first paragraph of section 2.2. However, we have updated our analysis with a revised comparison using PNOA-ALS validation data; therefore, we have removed the L4A analysis from the study.

P5, L122: The second approach seems more like a comparison rather than an actual methodology. What is your ground truth here?

Actually, GEDI L4A was a comparison method and a dataset for the subsequent Random Forest prediction. But, due to robustness lack we have decided to remove L4A data from the analysis. Also, we have understood that using GEDI L4A AGBD as a target variable is flawed in the olive orchard context.

P5, L123-124: Why integrate with remote sensing data? If it's for creating wall-to-wall maps, this isn't clear. Could you explain the reasoning?

It is integrated with remotely sensed data for creating wall-to-wall mapping. The manuscript indicates that it is also used to determine the importance of each feature. This has been referred in the manuscript to also referring to mapping:

"Remote sensing data provides a continuous spatial grid, while GEDI data provides data only at certain points at the footprint level. It is for this reason that we are assessing the ability of common remotely sensed data to reliably estimate biophysical variables by integrating both types of data". (lines 160-163)

P5, L125: At this stage, it's still unclear what optical and SAR data you are using. Could you clarify?

This paragraph is just an introduction to the methodological framework. In 2.3 section (data sources and Preprocessing) all data sources are explained in detail.

P7, L147: You still haven't explained why you are linking this to remote sensing data. Could you provide more context?

Yes, given this suggestion we have included the above-mentioned lines 160-163.

P7, L149: This is confusing. Are you using Random Forest to scale up biomass estimates, or for variable importance? Please clarify.

Yes, it is confusing because we assumed Random Forest is used to large scale estimations, also, RF was employed to assess the feature contributions. In the manuscript was modified:

"The Random Forest (RF) regression model is employed to large-scale predictions in the GEE platform and to analyze the contribution of each predictor variable, optimizing biomass prediction accuracy". (lines 163-164)

P7, L151-153: Delete the last sentence as it doesn't seem to add value.

Yes, this sentence doesn't add value to this section

P7, L163-165: This is not correct. I suggest reviewing how the L4A product is actually derived, as this part is misleading.

Yes, it was not correctly explained, but the GEDI L4A is removed from the original manuscript version, so we have removed it.

P7, L167: You can compare against L4A, but you cannot use it for validation. Validation implies true biomass measurements, which are not available here.

We agree with this point, so we have implemented a more reliable method to compare (PNOA- ALS).

P7, L167-169: This is incorrect. The GEDI L4A product does not directly relate GEDI L2A metrics to field-measured biomass. Instead, it uses a model inversion approach based on airborne LiDAR-derived AGBD estimates, which were previously calibrated with field plot data.

Yes, it was incorrectly explained. We have reviewed the L4A documentation, and the explanation was wrong. In any case, the L4A method is not used to validate.

P8, L181-182: Why use SAVI and BSI? Are they known to correlate with olive tree biomass? Could you provide more justification for this choice?

We are using SAVI index because is a Vegetation Index that we can control because the 'L' factor is regulated by the bare soil influence. It is developed because in Spain

there are multiple olive orchards conditions: eg. Orchards with grass above ground, without grass…

We have provided a more justified and arranged explanation in the new manuscript version in a new section: 2.4.1 Tree Density and Canopy Cover Modeling

P8, L183: Could you clarify the resolution at which you're processing the HLS imagery?

Yes, the resolution is 30m. It is said in the original manuscript. We have added a sentence that explains that it is the key layer to harmonize all multisource layers, eg: line 243.

P8, L184: Ground truth for what exactly? Is this intended for tree cover estimation? This needs to be clarified.

This expression was removed, this paragraph is confusing, and it was rewritten. Ground truth was referred to as the truth in the field, in any case, the full section is rewritten in the 2.4.1 section.

P8, L184-185: This is confusing. Is your canopy cover product based on aerial imagery or Sentinel-2 data? This needs to be clearer.

This sentence was removed. The new manuscript version includes a new section for explaining this process (Section 2.4.1).

P8, L189: Why PALSAR and not Sentinel-1? Can you explain your choice of SAR data?

It is a good question. We have many doubts about implementing Sentinel 1 in our dataset instead of PALSAR. However, ALOS PALSAR2 is a global 25m ALOS/PALSAR mosaic is a seamless global SAR image created by mosaicking strips of SAR imagery from PALSAR/PALSAR-2.

This is very useful for us because the filtering we are applying to the optical images includes filtering all images from the same year (applying a prior quality filter) and averaging all valid pixels. Therefore, PALSAR2 already implements this in the mosaic.

Furthermore, the L band is a very valid option for capturing the entire vertical structure of vegetation, while the C band is more sensitive to seasonal and temporal changes.

It has been mentioned in the manuscript: "PALSAR-2 is a highly valuable data source, as its operating frequency in the L-band allows for a more effective characterization of overall vegetation structure compared to other SAR systems operating in different frequency bands. For instance, C-band sensors offer higher spatial resolution but are more sensitive to seasonal variations (Qin et al., 2025)". (lines 204-207)

However, in future research, it would be very interesting to asses Sentinel 1's capacity for crop biomass estimations

P8, L204: Where are the details of canopy cover prediction? How do you distinguish olive tree canopy cover from other types of cover? This needs more explanation.

We agree this section needs a deep explanation. The new manuscript version includes a new section that explains this method in detail (Section 2.4.1: Tree Density and Canopy Cover Modeling). Also, Figure 2 illustrates the sampling procedure for training these subproducts.

P8, L206: Given the positional accuracy of the GEDI 25m footprint (~10m), does it make sense to use it at this scale? Could you clarify?

Yes, no doubt it is a substantial problem. The positional accuracy of each footprint is a relevant error source. However, the filtering by orchards greater than 20 hectares we think is a good strategy to mitigate this error, because, commonly, large orchards are ordered structurally by the same type of trees (species, age, agricultural conditions…).

Nevertheless, 10 meters of approximately horizontal accuracy within a pixel of 30m we think is a non-critical spatial accuracy problem.

P9, L215: Please clarify that crown diameter (Cdiam) refers to the average crown diameter per tree within the footprint. This needs to be more explicit.

Yes, maybe it is not clear. The new version of the manuscript includes this sentence to be more explicit.

"The crown diameter (Cdiam) is the average crown diameter per tree within the 30m pixel". (lines 277-278)

P9, L218: Olive tree heights range from 3m to 8m, and the RMSE of the L2A height product is about 5m. This creates a significant discrepancy. How do you address this uncertainty in tree height?

Yes, it is a critical issue. The new version of the manuscript includes an assessment of GEDI top height compared against PNOA high resolution. We have found relevant similarities between both data sources. So, we think it is not an important issue.

It is also commented in the discussion section.

P9, L227-229: This needs to be introduced earlier in the paper. How exactly have you derived tree numbers? What ground truth and predictors did you use for your supervised learning?

We agree it needs to be introduced earlier. The new version of the manuscript includes a section that explains the number of trees prediction and canopy cover prediction (Section 2.4.1).

P9, L235-236: L4A is not your approach; you are simply using it for comparison. Could you rephrase this section?

L4A was removed of this study, so rephrasing this section is not necessary.

P10, L248-249: What do you mean by "improve"? Are you using olive trees identified in high-resolution imagery to train a model for olive canopy cover prediction in HLS data?

Yes, correct. The 263 samples were used to create a linear model for estimating olive tree canopy cover prediction at 30m grid scale, using HLS (optical and infrared data).

P10, L258: I don't think the positional accuracy of GEDI L2A is suited for this type of work. Could you address this limitation?

We agree that the positional accuracy of GEDI L2A is an important issue to address. However, as we have explained before, the positioning variability is assumed as an error factor. However, we believe this error is minimized, since moving from the footprint level (~25m) to the pixel level (30m) already assumes this uncertainty in spatial adjustment. Therefore, we acknowledge this error and will discuss it in the discussion section.

P10, L272: I'm confused. Are you using both L2A and L4A as predictors? What's your response variable? Or are you using your volumetric approach (based on L2A and canopy cover) as the response? If so, this creates issues with data leakage, as you cannot use L2A as both the predictor and the response.

In the previous version of the manuscript, two approaches were proposed for large-scale mapping using Random Forest (this is distinct from the volumetric modeling stage): (i) a training dataset created with biomass data derived from the L2A volumetric approach (the main focus of the work), and (ii) a training dataset created with biomass values from L4A (directly AGBD L4A).

Regarding your question, AGBD L2A and L4A were not used as predictors. Table 2 details the data combinations used as predictors for each of the models. None of the models evaluated included L4A and L2A as predictors.

Of course, in the new version of the manuscript we will remove approach (ii), because we have come to the conclusion that in this work it leads to important confusion for two fundamental reasons: 1) The AGBD generated by L4A is not adapted to olive cultivation and it is not reliable to use it as a training dataset, and 2) It is not a robust validation/comparison method.

P11, Table 1: What are the predictor and response variables here? This table needs clarification.

Table 1 just shows all the remote sensing data, nevertheless, Table 2 explain in detail which are the predictors for each remote sensing multi-source combination

P13, L321: What's RVI? Please define this abbreviation clearly.

It is Radar Vegetation Index (RVI), it is defined below the Table 2, where all abbreviations are explained

P14, Figure 3: The frequency of GEDI L2A estimates is 10 times less than that of L4A. Why is this the case? Could you clarify this discrepancy?

Yes, it is a good observation that is not mentioned in the text. It is because the GPR includes a reliability factor in the predictions. We filtered the less reliable registers to obtain a better dataset. In the revised manuscript it is modified, because we decided to use RF instead of GPR to calculate 'tree density per hectare product'.

Also, we have recalculated all results because we decided to train the model again with a balanced dataset (Frequencies in the histogram of Figure 3).

L4A data doesn't include this kind of filtering, so the total register numbers are greater than the L2A volumetric method

P15: L355: In this section, it's unclear what exactly you're comparing and what your ground truth is. I assume you are comparing L2A-derived AGBD predicted using remote sensing data against the L4A AGBD product. However, this comparison is problematic without true biomass measurements for validation.

Yes, we agree it can be confusing. Actually, this comparison is showing the GEDI footprints that will be harmonized to create the training dataset. So, Figure 3 (panel 'a') shows all footprints for L4A, with itself histogram. Figure 3 (panel 'b') shows the total footprints used in this GEDI L2A volume-based approach, as well, with the histogram.

Note that GEDI L2A has a significant low footprints after removing unreliable values (explained in the previous comment).

The new version of the manuscript includes the PNOA-LiDAR approach, which will replace the L4A AGBD. It reduces substantially the number of validation points because in GEDI L4A we had had more than 100.000 footprints, but, in this case, PNOA is a supervised sampling method.

P16, Figure 4: You're comparing two highly inaccurate models. Without field measurements, this comparison is not meaningful. I suggest reconsidering this analysis

The new version of the manuscript includes the validation with PNOA approach, in order to rely on the validation dataset.

Technical corrections:

P4, L107: Consider using "key producer" instead of "key reference"

We considered "key reference" as an expression that includes the topics as: fruit production, spatial cover at large-scale, the sector for fruit transformation…

We would prefer to keep using "key reference"

P4, L107: Consider replacing "geometric normalization" with "co-registration", which is the more appropriate term in this context.

Yes, we agree this expression is more appropriate, because the method to align all datasets is the nearest neighbor resampling method. So, at the end, the original values are not changing when the alignment is done.

**Referee 2:**

The authors of this paper propose an integrative methodology to determine above-ground biomass density (AGBD) in olive orchards. The combined use of height data from GEDI L2A, SAR images, DEM, optical and infrared images can provide a lot of information on biomass distribution, especially in the case of olive orchards, a species of great agricultural and cultural importance in the countries of the Mediterranean basin. The validation and comparison of the results obtained with those modelled from other LiDAR data, in particular GEDI L4A, can be used to analyse the reliability of the proposed methodology. The analysis of the different regression models obtained with the different groups of variables used allows to identify the most important explanatory variables, although the grouping of all of them proves to be the best solution.

However, this ambitious goal has some weaknesses that need to be improved in order to make both the methodology and the analysis of the results clearer and more comprehensible:

- The explanation of the theoretical framework for biomass modelling is confusing and somewhat disorganised. Not enough details are provided for some parameters, e.g. canopy cover metrics or stem volume. How many field samples are used to train the Gaussian Process Regression that determines the total number of trees within each GEDI footprint? How is the number of trees per GEDI footprint determined from the size of the trees per hectare? I think more explanation is needed.

We agree that the theoretical framework for biomass modelling was disorganised, which is improved in the new version of the manuscript. We have added a new section to explain the tree density and canopy cover predictions (Section 2.4.1), and in the results, we explain the accuracy of these predictive models (Figure 3). We have reworked Figure 3, which shows the characteristics of the training data and the uncertainty associated with the GEDI height metrics, and both subproducts (Canopy Cover and Tree Density).

The canopy cover metrics were derived using a regression model trained to estimate canopy cover within specific GEDI footprints. These metrics were obtained from high-resolution PNOA imagery, incorporating spectral indices such as SAVI and BSI.

We acknowledge that this process was not thoroughly explained in the original manuscript, potentially making it difficult to follow. In the revised version, we provide a detailed description of the theoretical framework for biomass modelling, including how these metrics are computed (Section 2.4.1 and 2.4.2)

- This paper evaluates AGBD obtained by integrating data from different sources (with their limitations and accuracies) with the results of a model not specifically calibrated for olive trees (AGBD L4A), which makes the results less reliable. If validation were carried out against a ground truth, the estimates obtained for AGBD would be more reliable.

We fully agree that the lack of direct ground-truth validation represents a significant limitation of this study. In the revised manuscript, we have replaced GEDI L4A AGBD predictions with high-resolution ALS data from PNOA as the validation reference. This approach will allow us to directly compare volumetric estimates

derived from GEDI metrics with those derived from PNOA LiDAR. Subsequent validation was performed using the ground points derived by ALS-PNOA, compared against prediction carried out with GEDI L2A volumetric approach, after mapping at large scale.

We are aware that this approach also carries assumptions (namely, uniform wood density, fixed allometric parameters, and consistent stem-to-crown height ratios) which may not reflect local orchard heterogeneity. Nonetheless, the integration of ALS data as a validation source partly mitigates the absence of field biomass data. ALS from PNOA offers spatial precision and vertical accuracy offering a robust comparative benchmark, reinforcing the methodological soundness of our GEDI-derived predictions.

These issues have been added and improved in sections 2.6 of "Methodology" and sections 3.1 and 3.2 of "Results". In addition, the "Discussion" section has been expanded in an effort to improve the interpretation of these outcomes.

- Another important aspect of the work is the resolution of the GEDI footprints (25x25 m2) and their positional and vertical accuracy. In the 625 m2 of the GEDI footprint, there are very different types of olive trees, with regular and irregular planting frames and different stages of development, making it impossible to capture the fine-scale variations inherent to the crop. Other products have clearly better resolutions and accuracies, which should be detailed and justified why some are used and not others, such as the use of SRTM DEM (10m RMSE vertical accuracy) and not Copernicus DEM GLO-30 (2-4m RMSE vertical accuracy).

We appreciate this observation. To reduce variability caused by heterogeneity within GEDI footprints, we filtered out plots smaller than 20 ha, as large orchards tend to exhibit more uniform planting structures focused on fruit production. While irregular planting exists, it is less common in these large-scale commercial orchards.

We recognize that GEDI may not capture fine-scale vegetation structure optimally. However, for the scale and operational scope of this study, GEDI provides useful vertical structure metrics that are otherwise unavailable at large scales.

Regarding the selection of DEM data, while Copernicus DEM GLO-30 may offer higher accuracy, the SRTM DEM was chosen for its extensive use and ease of integration.

- In some parts of the methodology it is not entirely clear whether the authors have done some pre-processing of the data or whether they have accessed this information already pre-processed. There are no details of such pre-processing or the parameters that may have been used, so this should be clarified.

This is an important point. Most datasets used were already pre-processed to some degree. Table 1 ("Summary of Remote Sensing Variables and Features Used for AGBD Estimation in Olive Orchards") summarizes the derived products.

To clarify:

- HLS: Provided at surface reflectance level. Derived indices (e.g., SAVI, NDVI, BSI) are listed in Table 1.
- ALOS2-PALSAR2: Provided in DN. We converted to dB using:

$$\gamma_0 = 10 \cdot \log_{10}(DN^2) - 83.0 \text{ dB}.$$

Terrain correction was also applied, accounting for slope and incidence angle.

- SRTM DEM: Used directly in meters, that is already processed.
- GEDI: We extracted Relative Height metrics (RH95) from L2A and AGBD estimates and standard errors from L4A.

We clarify these steps in the updated manuscript and make the data processing chain more explicit in sections 2.3.1 - 2.3.3.

- It is not necessary to explain each acronym that appears at the caption of each figure or table, because once a term appears in the text for the first time, it is not advisable to repeat it each time.

On the other hand, tables and figures should ideally be self-explanatory, allowing readers to understand their content without needing to refer back to the text. While I appreciate the referee's perspective, we would suggest that acronyms should only be used if they are widely recognized and understood within the field

We have revised this issue in the manuscript to avoid repetitive expressions in the text.

- The temporal factor is not clearly detailed and analysed, i.e. what range of data is used to implement the methodology in Figure 2. Although the olive tree is an evergreen plant, the spectral variables obtained from HLS are highly conditioned by the dates analysed and the existing environmental conditions.

We acknowledge this point and have improved the temporal explanation in the revised manuscript. For each dataset used, we explain the temporal ranges used for each image.

For HLS data from 2020–2022, we applied cloud masking and then used the median of all valid pixels to synthesize spectral information into a single annual layer per year via Google Earth Engine (GEE). (lines 190-193)

For the SAR dataset (ALOS2-PALSAR2), no seasonal filtering was applied as the product is a global 25 m mosaic derived from multiple SAR scenes. we have expanded on these temporal considerations to make them clearer. (lines 204-2010)

- There is some confusion as to whether L2A and L4A are used as predictor variables, or whether one is used as predictor (L2A) and the other for validation (L4A).

We recognize that this part of the methodology may have caused confusion. In the revised manuscript, we restructure the workflow to eliminate misunderstanding.

Specifically, we validate GEDI L2A height-derived predictions using aerial PNOA LiDAR data instead of GEDI L4A. This change improves the reliability of the validation step and avoids the use of GEDI L4A product to validate another GEDI L2A volumetric approach.

In the original study, 16 Random Forest models were trained: 8 using L2A and 8 using L4A data, mainly to explore variable importance and point cloud distribution. However, this dual approach may obscure the study's primary intent. We will streamline the revised version to focus on validating GEDI L2A data with external ground-truth from LiDAR.

This new validation framework includes:

- Selecting representative sites
- Processing PNOA LiDAR to extract canopy height and cover

- Resampling to 30 m resolution to align with multisource satellite data

- There are some important errors in the citations to previous work: The data used or the results obtained and analysed are not as stated in this paper (see details in the specific comments).

We have revised these errors, and we have tried solving all of them.

In conclusion, I think it is an interesting paper with a lot of potential, but it needs significant improvements to be understandable, reliable and reproducible.

SPECIFIC COMMENTS:

Figure 1, understood as a graphical description of the working area, should not contain the results of the article. In any case, it could contain existing information in land use databases such as Corine Land Cover or SIGPAC (for Spain), but never results. The AGBD map obtained in this work for the whole of Europe should be included as a result in the corresponding section, as is the case for Spain.

The AGBD map has been moved to the results section, which is more appropriate in this case. We took the decision to set this map in study area to economize space in the manuscript.

In the table in Figure 1, the value of millions of hectares of olive groves in Spain is very different from the value given in the text (line 98), which was obtained from SIGPAC. This discrepancy should be explained.

Yes, it is a really good observation. It is because the SIGPAC database and CLC database have different precision in terms of delimitation accuracy.(Lines 105-109)

On one hand, SIGPAC is the plot boundary reference because it represents exactly the bounds of all olive orchard plots in Spain. On other hand, CLC is a European database that includes the class "Olive Groves", these bounds are less accurate than SIGPAC because CLC includes other land uses.

SIGPAC reports 2,695,055 hectares, meanwhile CLC reports 3,587,000 hectares. Note that for large scale estimation in GEE, the selected database was CLC due to better spatial cover.

Table 1 is referred to in the text on line 181, but is located in line 277, it is quite separate from the text, which makes its interpretation somewhat more complex. In this table, the detail of all the variables used at the bottom of the table is somewhat excessive when it could be placed in the text, for example on line 185.

Yes, the Table was located on this site to economize some space, but it could indeed make it difficult the correctly understand the feature selection. So we locate this table earlier in the new manuscript version.

The citation Velàzquez-Martì et al. has a spelling error in the accents, which are repeated each time it appears, and should be Velázquez-Martí et al.

Yes, it is indeed corrected in the new manuscript version

Line 33: The natural process of carbon storage by olive trees contributes to reducing the amount of greenhouse gases in the air, but does not reduce greenhouse gas emissions per se.

Yes, it is rephrased in the new manuscript version:

"This natural process helps mitigate climate change by reducing the concentration of greenhouse gases in the atmosphere (Pardo et al., 2017)" (line 34)

Line 55: This part of the introduction talks about biomass in olive crops, but line 55 cites work related to forest biomass. I think the introduction could be reorganised to emphasise the importance of work focused on forest environments and to highlight the differences with the analysis of olive tree biomass.

Yes, it could be reorganized. The current version emphasizes the approaches used in remote sensing to estimate biomass, highlighting the labor cost in sampling.

Line 125: GEE is an environment for accessing, not acquiring, satellite imagery, among other products or data. Acquisition of satellite imagery has other connotations.

Yes, acquisition could have other connotations, so we rephrased this section

Line 144: The various datasets are said to be pre-processed, including atmospheric corrections and geometric normalisation. The article gives no details of this pre-processing, nor of the algorithms or parameters used. Line 177 indicates that the HLS product is pre-processed, so there is some contradiction with line 144. I think it is very important to clarify how far the authors have gone in their work on pre-processing and, if it has been done, how it has been done.

This line was deleted because there was a contradiction, as you reported. The pre-processing is explained in section 2.3 (Data Sources and Preprocessing) for each data source.

Line 146: I'm not sure I understand the sentence between lines 146 and 148.

This sentence expresses the improvement that suggests using SIGPAC data for building the training dataset. It allows relying on the location and quality dataset. The sentence has been modified:

 "To improve spatial accuracy in olive grove delineation, SIGPAC cadastral data is used to refine the initial classification from the CLC database, that is, SIGPAC was used for the filtering of selected sites for training and testing". (lines 154-156)

Lines 148 and 149: This sentence does not say the same as it appears in Figure 1.

Yes, this sentence has been rewritten:

"L2A canopy height metrics serves as canopy height values for AGBD estimation after its processing using a volumetric approach adapted to olive cultivation". (lines 158-159)

Line 156: It is necessary to specify the characteristics of the GEDI LiDAR sensor, its resolutions, accuracy, etc., as well as its products L2A and L4A (line 165).

Yes, we have completed this section in the new version of the manuscript, specifying the limitations about positional accuracy and vertical metrics. We think it is important to comment about this issue, because it is one of the critical issues of this study. (lines 175-176)

Line 166: It is stated "AGBD values were derived from L4A using ..." but in Figure 2 it is stated that these values are part of the data accessible in GEE. Regarding this variable, lines 259 and 272 refer to it as a 'predictor variable' in the models, but in Figure 1 it appears to be used only to compare and validate the model that estimates

AGBD. I find this confusing and do not clearly understand the role of AGBD L4A in the workflow.

Yes, it is true. We think this is the more critical point, in the first version, because, in the workflow (Figure 1), it was not clearly expressed. So, we have decided to remove this predictive approach to avoid misunderstandings.

The current Figure workflow shows more clearly all the processes developed along the study.

So, we have kept the GEDI L2A volumetric approach and the derived biomass product (AGBD) as the training dataset. The L4A was removed from the study.

Line 188: Details of how the global annual mosaic was generated and the pre-processing applied to the PALSAR images are missing. Why were these images used rather than Sentinel-1, which has better spatial resolution? PALSAR operates in L-band, with greater penetration of vegetation canopies than Sentinel-1 (C-band), but for small canopies (olive trees) it is not clear that this is an advantage over the greater spatial detail that can be obtained with Sentinel-1.

The mosaic is already generated in the product that GEE offers, and the pre-processing is explained in the new manuscript version. (lines 208-2010)

Regarding the justification of using PALSAR instead of Sentinel 1. Probably, for accuracy, Sentinel 1 (C-band) could be a better option than PALSAR. However, for work purposes, we decided to use PALSAR. Also, the resolution of these images is closer to 30m of HLS (key layer).

As previously mentioned , for located studies with other purposes, Sentinel 1 can be a more reliable option.

Line 196: Why SRTM (10 m RMSE vertical accuracy) is used rather than other global models with higher accuracy such as Copernicus DEM GLO-30 (2-4 m RMSE vertical accuracy), which is also much more recent (2011-2015) than SRTM (2000)?

It was very important for us that the dataset be included in the GEE catalog so that the model would be easily reproducible. This product was not used because, when the framework and datasets were being designed, we were unaware of its existence within GEE.

We will consider it for future research; however, SRTM is a widely used and proven dataset.

Lines 207 to 209: the same citation is used to indicate the value used in a variable (WD) and to justify or validate its use. For the latter, the citation from Velázquez-Martí 2014 cannot be included.

Yes, Velázquez-Martí et al., (2014) was removed in the validation of the use of the constant WD.

Line 241: Has a minimum threshold of olive trees been used to apply this selection or filter? The GEDI footprints (25x25 m2) have to be within the olive tree plots, so it is important to use an excess threshold for this to happen and to be able to apply erosion filters (line 253).

We are not sure if we understand this question correctly. We are going to explain the procedure for filtering by olive orchard plots:

1) For all SIGPAC polygons across Spain we filtered by orchards greater than 20 hectares. We did it to avoid small plots that could contaminate the train dataset because in a small plot the footprint (~25m diameter) could be between two different land uses.

2) We rasterized polygons over 20 hectares (previously filtered) and we applied an erode filter. We did it to avoid edge effects in the footprint sampling; therefore, we selected footprints fully within the olive orchard plots.

We hope it solves your question. So, the erode filter was applied to selected plots, not applied directly to any minimum threshold referred to olive trees within the footprint, because we assume that 25m diameter will always contain at least one tree. The procedure previously explained is in lines 305-309.

From line 295 to 305 the idea of using different combinations of variables to estimate biomass is repeated three times. These paragraphs should be rewritten to avoid repetition of ideas.

Yes, we have rephrased these sentences to avoid repetition of ideas and be more fluent: The model validation was conducted through a cross-validation process using the test dataset, which represented the totally PNOA-ALS points sampled. Standard performance metrics, including the coefficient of determination ($R^2$) and root mean square error (RMSE), were calculated. To enhance the robustness of biomass estimation, various input variables derived from optical, radar, and topographic datasets (Table 2) were systematically evaluated. This approach allowed for a comprehensive assessment of feature importance and the individual contribution of each geospatial data source.

Line 329: The last paragraph of section 2.6 looks like a conclusion, but it is in the data and methodology section.

In this last paragraph we have removed the sentence: 'The spatially explicit biomass estimates support biomass monitoring, carbon sequestration assessments, and sustainable agricultural management in Mediterranean landscapes '

Line 339, Figure 3. The size of the dot symbol looks different in (a) and (b), causing confusion. The provincial boundaries (with different colours) are not relevant and do not add anything as there is no spatial analysis by province. In any case, since they are mentioned in the text (line 336), the Autonomous Communities could be added. And the symbol used in the legend should be included.

Regarding the size of dot symbols, it is a good observation, but it is a problem in the creation of the map. However, for the new version of the manuscript Figure 3 (panel a)has disappeared because GEDI l4a is removed from the analysis.

About the provincial boundaries, indeed, it is not showing any spatial analysis with these bounds. We can change the spatial bounds to Autonomous Communities for a better comprehension. We have added these bounds in the map legend (Figure 3).

Line 344: I don't understand that after filtering the GEDI prints by olive tree parcels, you are now talking about cover crops such as grasslands, shrubs and woods, and they appear in the results.

Yes, maybe it is not correctly explained. This part of the results is referred to the results of GEDI L4A, which, this processing level of the GEDI product applies different models to the footprints. Actually, the footprints are filtered by the olive

orchards plots, but GEDI L4A (AGBD) doesn't take it in account. So L4A is processing these footprints as a other kind of vegetation, eg: grasslands, shrubs. Although in reality, these footprints correspond to land use "olive trees", that GEDI l4a doesn't cover this kind of land use.

Nevertheless, GEDI L4A has been removed from the analysis, so this is removed from the manuscript.

Line 357: Mg/ha is given as the unit of measurement, whereas in other cases Ton/ha is used. It should be homogenised.

We agree, it has been homogenized in the revised manuscript

Line 369: 0.30 should be corrected to 0.29.

Yes, it should be corrected to 0.29, as the Figure 4 reports. However, this analysis is referred to L4A.

Line 372 can be combined with line 373 to give continuity of meaning.

It will be modified because Plant Functional Types is linked with L4A.

Lines 384 to 387: this statement is repeated several times and seems to be a conclusion rather than a result in itself.

Yes, it is true, this paragraph is like a brief conclusion of the results. It doesn't show any results. However, we prefer to leave this paragraph to emphasize the fully multisource model and summarize this section a little.

Line 390 to 392: Combine these two sentences to make it clear that Andalusia is in the south of Spain.

Yes, we have combined these sentences to inform the location of the Andalusia region:

'The spatial distribution of AGBD across Mediterranean olive-growing regions exhibits significant variations at both national and sub-national levels. Spain presents the highest total AGB among the analyzed countries, with extensive areas of high-density biomass predominantly located in Andalusia, a region in southern Spain. The highest biomass values, exceeding 50 Mg·ha⁻¹, are concentrated in this southern part of the country, particularly in regions characterized by intensive olive cultivation. Other areas in Spain, such as Castilla-La Mancha and Catalonia, display moderate AGBD values, ranging between 20 and 40 Mg·ha⁻¹, while lower biomass densities, below 15 Mg·ha⁻¹, are observed in more marginal olive-growing areas (Figure 5)'. (lines 448-454)

Line 407: "AGBD mean" should be deleted.

Yes, it was a writing error.

Section 3.4. The way it is written, it could be part of the discussion because it does not give numerical results, but an analysis and interpretation of results that are not there. In this sense, something is said about the most influential predictor variables within each group of variables, but there are no models, values, weights, explanatory percentages or importance of each variable individually to confirm that NDVI and NDWI are the most influential variables, for instance.

This part has been rewritten to include some numerical results, also, we have written this section avoiding results interpretations.

Line 445: the paper by Estornell et al. 2015 does not use spectral reflectance but LiDAR variables.

You're right; this citation was incorrect. The intended reference was:

Estornell, J., Ruiz, L. A., Velázquez-Martí, B., and Hermosilla, T.: Estimation of biomass and volume of shrub vegetation using LiDAR and spectral data in a Mediterranean environment, Biomass Bioenergy, 46, 710–721, https://doi.org/10.1016/j.biombioe.2012.06.023, 2012.

We will correct it accordingly.

Line 492: The work by Fernández-Sarría et al. 2019 does not compare olive tree biomass with forest biomass, but only studies the residual biomass from olive tree pruning, logically with its structural planting framework.

Yes, we agree that this reference is wrong, we have removed it.

Line 501: UAVs were not used in either of the cited studies: aerial LiDAR was used in the 2015 study and TLS in the 2019 study.

Yes, it is true. The reference was meant to highlight the use of high-resolution LiDAR (aerial or terrestrial). We have rephrased this sentence:

"While high resolution LiDAR approaches have demonstrated high precision in small-scale biomass assessments (Estornell et al., 2012; Fernández-Sarría et al., 2019)".
(lines 581-582)

Row 529: The cited papers do not analyse irrigation or soil types or how they may affect biomass estimates. And only one dataset (TLS) is used. The statement "highlights the need for more comprehensive datasets in future research" is very general and cannot be limited to this work.

Yes, the cited papers have been removed.

Line 567: This conclusion is too optimistic. With an R2 of 0.62 and an RMSE of 5.95 Ton/ha, it cannot be said that the model successfully captures the spatial heterogeneity of biomass, and even less so for the whole Mediterranean. This R2 is not high and the reported RMSE is higher than desirable.

We agree in the point that an R2 of 0.62 is not high to determine the total heterogeneity of olive orchards across Mediterranean. However, taking into account the challenging task of biomass estimations, and the complexity of the predictable variable, we think it is a positive result. Especially given the predictors used, fully remote sensing variables provided by satellite imagery.

In the revised manuscript, the results have been recalculated using a more balanced training dataset to mitigate the tendency of the Random Forest model to underestimate biomass values. While this adjustment has slightly reduced the model's overall performance ($R^2 = 0.56$ with respect to the modeled validation dataset), the comparison against reference data yields a considerably lower $R^2 = 0.14$, indicating limited precision. However, we consider these results promising given the complexity and scale of the study area, as well as the constraints inherent to remote sensing-based biomass estimation. Nevertheless, we acknowledge that the current approach is not well-suited for precise assessments at the farm level or for detailed yield monitoring in

individual plots. For such applications, methodological refinements and more accurate input data would be necessary to improve reliability.

Nevertheless, have rephrased this section to avoid overinterpretation of the model performance.

---

## Referee Report (RR1)

The authors of the paper have done a good job of taking the two reviewers' suggestions from the first round of reviews into consideration. They have made important improvements that enhance the quality and clarity of the work. Nevertheless, I believe it is necessary to make some recommendations to clarify certain sections of the text.

Ideally, there would be a map of the study area, or at least a list of the countries to be analysed (those in Figure 5).

In the new methodology section, ALS data are used; a description of these can be found from line 319 onwards. These data should be described earlier, in section 2.3 Data sources and preprocessing, and the date should be indicated, as well as the implications that the date difference with the rest of the data could have.

Lines 131 and 132: The idea of estimating AGBD from GEDI L2A is repeated.

Line 273: 'was applied' is missing.

Line 323: ')' is missing.

---

## Author Response (AR2)

The authors would like to thank the editor and reviewers for their comments, which have helped improve the scientific quality and clarity of the manuscript. Below, we respond to the feedback from the editor and the reports provided by the reviewers. The responses are highlighted in blue.

Below, we respond to the **3 points** that the Editor suggested to improve in the revised manuscript version.

**1)** The quantitative results do not match the authors' conclusions described in the text. While I do think this method can be published in some form as an attempt to predict olive tree biomass, I think it is a much more negative result than the authors are expressing. Potentially, the paper should be about an estimation of the error or uncertainty of olive biomass with the best remote sensing tools we have and/or about how we currently don't have sufficient tools to predict this data at large scales. I'm not sure we can draw such confident conclusions from the analysis as stated in lines 19-20 and 475-476 for example, especially based on the results shown in the figures. The overarching statements that model performance up to R2 of 0.56 R2 in line 634 is misleading when showing several times lower performance values for independent lidar references (less than 0.14). This is described more in the second point. The lack of sufficient comparison with independent, reference LiDAR data is a major concern. It seems like the authors still express some confidence in the approach despite very low fit values, which is misleading to someone reading the abstract.

We fully agree with the reviewer's comment. In some sections of the original manuscript, the predictive model was described as providing a relatively accurate solution for biomass quantification in olive orchards. We have revised these parts of the text to avoid any potential misunderstanding or overstatement. Specifically, the sentences mentioned by the reviewer (lines 19–20 and 475–476) have been rewritten, along with other passages that could lead to confusion or be perceived as overly optimistic. All these changes are highlighted in the "Track-Change" document.

Regarding the R² value of 0.56 mentioned in the manuscript, this corresponds to a different analysis than the one performed for the validation with ALS data. That value refers to the internal predictive performance of the Random Forest model, evaluated

using a 25% validation subset derived from the volumetric modeling framework. This process is described in lines 138–139 and further detailed in lines 351–356. Initially, we considered removing this validation method. However, we concluded that while it may not be the most objective approach, it is useful for understanding RF's performance with internal model validation. We have therefore decided to retain it in the manuscript. Nonetheless, we emphasize that the ALS validation is the most valuable one in our study.

Additionally, we have clarified in the revised Discussion that the low correspondence with ALS data highlights the current limitations of GEDI and optical-radar inputs for low-stature vegetation. The revised text now emphasizes that the proposed framework should be interpreted as a methodological exploration rather than a fully accurate operational model.

**2)** The validation data as GEDI L2 data seems circular and/or insufficient for the application here. It appears the validation data still appear to be a version of GEDI which is highly circular considering it is used as one predictor. This issue was raised in the first round by both reviewers as a major concern. In my decision in the first round, I mentioned that this should be sufficiently addressed. However, the authors still use GEDI as the reference here and based their main conclusions mostly on these results, which does not appear appropriate (it appears to be both a regressor and reference data for training). In this next round, the new reviewer also brought up this issue, and I also agree this is a concern. Given consensus of concern from several reviewers, this issue needs to be addressed given that many readers will likely take issue with this. The main issue is that while the values are higher in the top part of figure 4, these are both error prone data to compare to and is a bit circular to have predictors be the same as predicted. Or if this is not the case that GEDI L2 data are both being used as reference and as a predictor, then it was still not clear what predicted and predictor data are being used (please note that it is still quite challenging to determine what the reference/validation data are). It ultimately needs to be more clearly motivated why this approach is chosen.

We agree that the validation approach in the first version of the manuscript was not sufficiently robust. For this reason, in the revised version we have included an independent validation using ALS data, which are free from the uncertainty associated

with GEDI-derived measurements. However, it is important to clarify that the validation section now includes two separate analyses:

1- Internal model validation using GEDI L2A-derived data (25% of the total training samples shown in Figure 3). We acknowledge the concern regarding potential data circularity, but we emphasize that the GEDI-based validation set was never used during model training. It was exclusively reserved to evaluate model performance using the predictor variables listed in Table 2. Importantly, none of these predictors include GEDI height metrics; GEDI L2A data and derived sub-products ('canopy cover' and 'Tree Density') were only used to construct the training dataset (i.e, as the source of reference biomass information), not as input variables for model prediction.

2- Independent model validation using ALS data. This represents the most reliable and independent validation dataset in our study. We have clarified throughout the revised manuscript that this validation provides the main quantitative benchmark of model performance.

In addition, several sentences have been reworded in the "Methodology" and "Results" sections to better distinguish these two analyses and to explicitly highlight the significance of the ALS-based validation as the definitive assessment of the model's predictive accuracy.

**3)** As the reviewers pointed out in the first round, and so does the new expert reviewer, GEDI's waveform approach will struggle with trees that are near the height of the noise level. The authors do not appear to sufficiently address this point in the text about GEDI's use here. As the new reviewer pointed out, the GEDI instrument's noise level is around a few meters and olive trees may be near this height. I do think it is possible for GEDI to detect these olive trees (especially if nearing 10m), but the uncertainty bar will be higher. This issue needs to be sufficiently addressed in this paper (especially as a major discussion point), and especially why this issue is not mitigated by use of other data sources (optical and active microwave).

We appreciate the reviewer's insightful comment, which raises a highly relevant concern that has significantly improved our analysis and discussion. In this revised version, we have analyzed the GEDI L2A training dataset by extracting the Relative

Height (RH) percentiles for all footprints and examining the internal distribution of canopy height metrics. A new Figure 4 has been added to illustrate this analysis: Figure 4a shows the mean relative height waveforms for different olive tree height classes, while Figure 4b displays the percentile at which the first RH value greater than zero occurs for all training points. This metric serves as a useful indicator of waveform noise and allows us to assess its variation across vegetation height classes.

This analysis confirms that waveform noise strongly affects lower vegetation and supports our decision not to use the GEDI L2B "Canopy Cover" product in the training dataset due to its high uncertainty for low-stature trees.

We have incorporated a detailed explanation of this analysis in Section "3.1. Training Phase: Biomass Modelling Framework for AGBD Inputs and Relative Height Metrics Assessment" (lines 424–439). In addition, a more extensive discussion of these findings has been added in Section "4.1. Model Performance and Data Integration" (lines 555–579), where we address the limitations of GEDI for detecting short canopies and the implications for biomass estimation in olive orchards.

Finally, we note that this limitation makes it challenging for the model to fully exploit the potential of optical and SAR variables within the current framework. This is because the AGBD modelling approach (as shown in Figure 1) relies exclusively on the three predictor variables derived from the volumetric model, which are themselves based on the GEDI reference dataset. Consequently, the training data are inherently biased toward GEDI-related structural information and subproducts ('Canopy Cover' and 'Tree Density'), making it difficult for the model to fully exploit the complementary potential of optical and radar variables. To fully benefit from multi-source integration, the entire modelling framework would need to be redesigned to incorporate these data sources more explicitly. It is explained in lines 706-711.

Finally, note that R2 values, by definition, can't be negative (in figure 4).

We appreciate this observation. By definition, the coefficient of determination ($R^2$) cannot take negative values when calculated as the square of the Pearson correlation coefficient. The negative value initially reported was due to the use of the "coefficient of determination" as returned by the python function "sklearn.metrics.r2_score",

which can produce negative values when the predictive model performs worse than the mean predictor.

**Anonymous Referee #2**

The authors of the paper have done a good job of taking the two reviewers' suggestions from the first round of reviews into consideration. They have made important improvements that enhance the quality and clarity of the work. Nevertheless, I believe it is necessary to make some recommendations to clarify certain sections of the text.

Ideally, there would be a map of the study area, or at least a list of the countries to be analysed (those in Figure 5).

We initially considered including a map or a detailed list of countries. However, to keep the manuscript concise and focused, we decided not to add additional figures. We believe that the current textual description of the study area provides sufficient contextual information for readers to interpret the analyses presented in Figure 5.

Furthermore, the table shown in Figure 6 contains all the countries analyzed, so we believe it is quite clear.

In the new methodology section, ALS data are used; a description of these can be found from line 319 onwards. These data should be described earlier, in section 2.3 Data sources and preprocessing, and the date should be indicated, as well as the implications that the date difference with the rest of the data could have.

We appreciate this constructive suggestion. In the revised version, we have added a new subsection titled "2.3.4. Airborne LiDAR Scanning (ALS) Data", where the ALS dataset is introduced earlier in the manuscript.

Lines 131 and 132: The idea of estimating AGBD from GEDI L2A is repeated.

Corrected: "The proposed framework derives AGBD estimates from the GEDI L2A product (Dubayah et al., 2021)".

Line 273: 'was applied' is missing.

Corrected: "This WD value was applied as a constant to convert volumes into biomass units."

Line 323: ')' is missing.

Corrected

**Anonymous Referee #3**

This study uses Random Forest method to predict the biomass of Mediterranean olive orchards with multi-source remote sensing data, including lidar, optical, and SAR datasets. The study goal is to determine the most effective combination of RS features for biomass estimation. The conclusion is to include all sources that were proposed in this study. However, I suggest rejecting this manuscript due to the following reasons:

• The height of olive orchard is generally smaller than 5m (Fig3.b). GEDI's waveform FWHM is about 5m. Olive orchard's height is below the limit of GEDI sensor. The heights of olive orchard in this study ranges from 2 to 5m. GEDI will not be sensitive to such small changes. This is also the reason why R2 is only 0.16 in the result.

We fully acknowledge this limitation. In this revised version, we have included additional analyses and discussion specifically addressing this issue (Figure 4). We agree that the relatively low canopy height of olive orchards, close to the GEDI sensor's detection threshold, is one of the main reasons for the reduced predictive accuracy observed. However, the main objective of this study is not to present a definitive model for biomass estimation in low-stature vegetation, but rather to provide a theoretical framework that evaluates the feasibility and limitations of using GEDI-derived height metrics for such ecosystems. In this sense, our results should be interpreted as an exploratory assessment that highlights both the current constraints and the potential for improvement with future advances in remote sensing instrumentation.

• Canopy cover is already product in GEDI L2B. The authors used a single index – SAVI to predict canopy cover. It doesn't make sense.

We understand that not using the GEDI L2B "Canopy Cover" product might seem counterintuitive. However, our analyses showed that this product performs poorly for low-stature vegetation such as olive orchards. In contrast, using a canopy cover estimate derived from spectral indices (specifically SAVI) provided more consistent results. Furthermore, our assessment of the GEDI waveforms, including the distribution of relative height percentiles, revealed significant inaccuracies in the L2B product for short vegetation canopies. These findings reinforced our decision to exclude the GEDI L2B "Canopy Cover" product and rely instead on alternative optical-based predictors better suited to this type of ecosystem.

• The value of AGBD is from 0 to 50 Mg/ha. The value is small, and variance is also small. It is difficult to tell the difference in model performance. In "Figure 1. Density

plots for the regression model AGBD estimation", the $R^2$ is only 0.14, also too low when validating with reference datasets.

The AGBD range may appear low compared to other forest species. However, the upper limit of 50 Mg/ha was selected because it represents a realistic threshold for olive orchards, as olive trees rarely exceed this biomass value per hectare. This range is consistent with previous studies and field-based assessments of olive biomass. The relatively small variance is therefore inherent to the structural characteristics of olive trees rather than a limitation of the dataset itself.

Regarding the low $R^2$ value, we fully acknowledge that the model performance is limited. Nevertheless, this study is intended as an exploratory assessment rather than a definitive biomass estimation approach. Its main goal is to evaluate the feasibility of using GEDI-derived height metrics within a volumetric modelling framework, to test different multi-source remote sensing combinations, and to discuss the inherent limitations of GEDI measurements over low-stature and cultivated vegetation.

I appreciate authors spent so much effort on this study, but I didn't agree that GEDI is good biomass estimation for Olive orchard's in this way.

We sincerely appreciate the reviewer's recognition. We fully understand the limitations of the GEDI sensor for olive orchards. However, despite these constraints, we believe the study provides valuable exploratory insights. It contributes to understanding the feasibility and current challenges of applying GEDI data for biomass estimation in cultivated and low-stature ecosystems, while highlighting areas for methodological improvement and future research.